# Climate and land management accelerate the Brazilian water cycle

**Vinícius B. P. Chagas** [1] ✉, **Pedro L. B. Chaffe** [2] ✉ **& Günter Blöschl** [3]

Increasing floods and droughts are raising concerns of an accelerating water cycle, however, the relative contributions to streamflow changes from climate and land management have not been assessed at the continental scale. We analyze streamflow data in major South American tropical river basins and show that water use and deforestation have amplified climate change effects on streamflow extremes over the past four decades. Drying (fewer floods and more droughts) is aligned with decreasing rainfall and increasing water use in agricultural zones and occurs in 42% of the study area. Acceleration (both more severe floods and droughts) is related to more extreme rainfall and deforestation and occurs in 29% of the study area, including southern Amazonia. The regionally accelerating water cycle may have adverse global impacts on carbon sequestration and food security.

Floods and droughts cause more damage worldwide than any other natural hazard[1,2] and their risks may be exacerbated by climate change and socio-economic activities[1,3,4]. Often an increase in floods is aligned with a decrease in droughts as a result of more abundant rainfall, and the opposite is the case as rainfall becomes scarcer[5,6]. However, some models suggest a joint increase in the severity of floods and droughts[1,7,8], a phenomenon referred to as acceleration of the terrestrial component of the water cycle. This acceleration could lead to large compound impacts[9] on global food production[10,11], ecosystem health[12,13], and infrastructure[8].

There are a number of processes that potentially cause an acceleration of the water cycle. In a warming climate, the moisture carrying capacity of the atmosphere is increased[14] enhancing extreme rainfall[15,16] which may increase streamflow during floods. Enhancement of rainfall seasonality[17] may decrease streamflow during hydrological droughts. Additionally, the global atmospheric and oceanic circulations are affected[7,14,18]. Weaker meridional pressure gradients in a warmer climate may lead to the amplification of stationary waves causing more persistent rainfall and drought periods[19] and rapid shifts between these two regimes[20–22]. Changes in monsoon patterns with increasing contrasts between land and sea surface temperature[7] can similarly increase floods and droughts. Land management can also accelerate the water cycle. Agricultural practices can reduce rainwater infiltration into the soil which increases overland flow and thus floods,

and reduces groundwater recharge and thus low flows during droughts[23]. River engineering[8], urbanization[24], and groundwater pumping[25] can have similar effects on streamflow. While there is some evidence for the acceleration of the water cycle over the ocean[18], there is little such evidence over land[26,27] because of insufficient streamflow data and the confounding effects of the growing human interference in the terrestrial water cycle[8].

Here, we analyze a comprehensive hydrometeorological, land cover, and human water use data set in Brazil and show that water use and deforestation have amplified climate change effects on Brazilian streamflow extremes over the past four decades. This region encompasses some of the world's largest basins with mounting concerns of changing floods and droughts[6]. Our analysis is based on daily streamflow observations from 886 hydrometric stations (Supplementary Fig. 1) for the period from 1980 to 2015. For each station, we compute annual time series of annual minimum 7-day streamflow as a measure of drought flows, mean daily streamflow as a measure of water availability, and annual maximum daily streamflow as a measure of flood flows. We quantify the trend magnitude of each time series (i.e., local trend) with the Theil-Sen slope estimator, the significance of each trend with the Mann-Kendall test, and obtain regional trends by spatial interpolation with ordinary kriging.

For each basin, we consider three climate drivers of streamflow change, computed from daily meteorological data from 1980 to 2015:

---

[1]Graduate Program of Environmental Engineering, Federal University of Santa Catarina, Florianopolis, Brazil. [2]Department of Sanitary and Environmental Engineering, Federal University of Santa Catarina, Florianopolis, Brazil. [3]Institute of Hydraulic Engineering and Water Resources Management, Technische Universität Wien, Vienna, Austria. ✉e-mail: vbchagas@gmail.com; pedro.chaffe@ufsc.br

(i) mean daily atmospheric water balance, computed as precipitation (P) minus evaporation (E, including transpiration from plants); (ii) annual minimum 90-day P − E to indicate droughts caused by seasonal variability; and (iii) annual maximum 14-day P − E because, as basin response times range from less than a day in small basins to a few months in large basins, the 14-day time scale is a compromise on which basins are most sensitive. Additionally, we consider two non-climatic drivers: (i) water use for irrigation and other purposes, and (ii) native vegetation cover. All variables are analyzed in units of mm d$^{-1}$ so that they are independent of basin size, except for native vegetation which is analyzed in % of basin area. Trends in streamflow and their climate drivers are expressed in units of % per decade by dividing each trend by the long-term average value of the same time series. We analyze the links between streamflow changes and their drivers with panel regressions, which allows us to investigate the hydrological variability in space and time in a single framework. We set the regressions with fixed effects for location and use logarithmic-transformed variables. In addition, we explore the acceleration of the water cycle with bivariate frequency distributions with respect to flood and drought flows.

## Results

### Detection and attribution of streamflow trends

Our data show that streamflow changes have been widespread in Brazil (Fig. 1). Further diminishing low flows (i.e., increasing severity of hydrological droughts) can be found in southern Amazonia and central-eastern Brazil (Fig. 1a) while increasing flood flows can be found in Amazonia and in the southeast (Fig. 1b). Regional trends in drought flows range from −37 to +16% per decade and those in flood flows from −17 to +10% per decade. Local trends in drought flows at the hydrometric stations (Supplementary Fig. 2) range from −65 to +59% per decade and those in flood flows from −39 to +32% per decade. The average trends over the entire domain are −5% and −1% per decade for droughts and floods, respectively. Out of the 886 stations, 353 and 56 stations show significantly (α = 0.05) decreasing and increasing drought flows, respectively, and the corresponding figures for floods are 104 and 51 stations.

The regression analysis suggests that streamflow change is related to the combined effects of climate variability and increasing water use

(Fig. 1c, d). Drought trends are driven primarily by changes in mean daily P − E, with substantial effects of water use and minimum P−E (Fig. 1c). Water use impacts are noticeable particularly in central-eastern Brazil, where decreases in drought flow and increases in water abstraction are the greatest (Supplementary Figs. 2–4). Flood changes are related to maximum P − E and mean daily P − E (Fig. 1d), indicating that the floods change in response to modified extreme precipitation and antecedent soil moisture conditions.

To interpret our results, we focus on four hotspots of change with distinct streamflow regimes, land management, and in the upstream areas of major South American basins with mounting environmental concerns such as the Amazon, São Francisco, Paraná, Uruguay and Iguaçu basins (Supplementary Figs. 2–4). In the southern Brazil and northern Amazonia hotspots, drought flows are aligned with increasing mean P − E and minimum P − E with little land management effect on streamflow (Fig. 2). Floods in the southern Brazil hotspot, a subtropical region, have increased in line with increasing maximum P − E and mean P − E. In the Highlands hotspot, a region with intensive agriculture, the reduction of drought flows is aligned with decreasing mean P − E and increasing water use but, from the year 2000 onward, drought flows have become dissociated from mean P − E with a rapid increase in water use (Supplementary Fig. 7). In the southern Amazonia hotspot, drought flows have decreased substantially, even though the climatic variables have barely changed, suggesting an effect of large-scale deforestation of the tropical rainforest.

### Four quadrants of streamflow change

Changes in the extremes may not always be synchronized with changes in mean flows. For example, an increase in mean streamflow combined with an increase in the variance of streamflow could lead to increasing high flows but decreasing low flows. Here, we examine how both flow extremes have changed in a single analysis by classifying the trends into four quadrants (Fig. 3a). The northern Amazonia and southern Brazil hotspots show increases in flood and drought flows (wetting conditions), which implies that floods have become more frequent and droughts less frequent. The Brazilian Highlands show decreasing flood and drought flows (drying), and southern Amazonia increasing floods and decreasing drought flows (accelerating). Even though the trends in

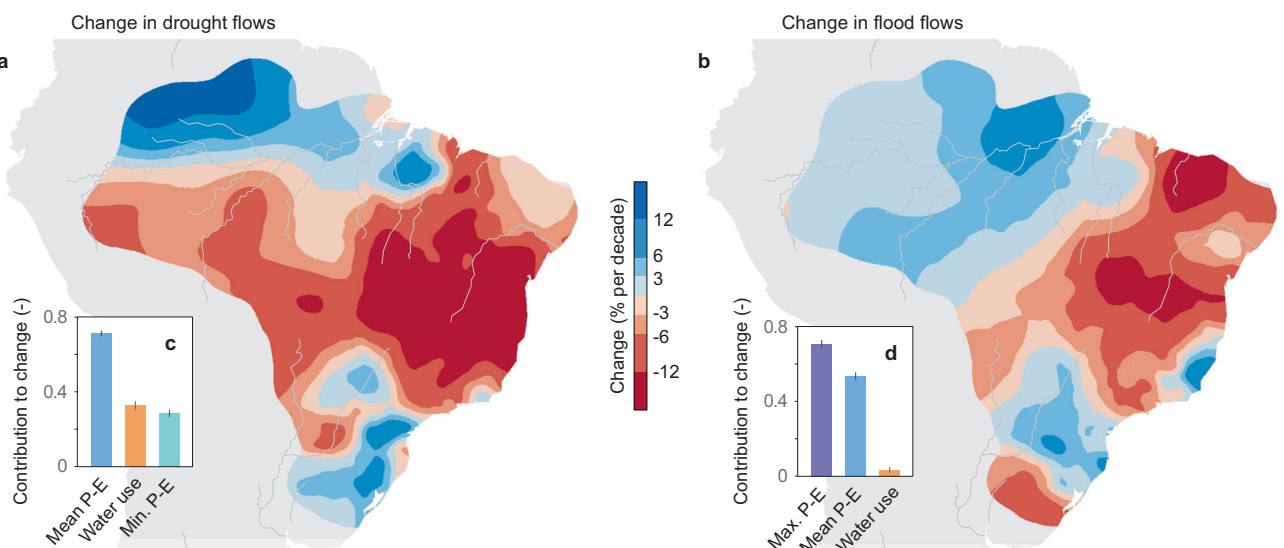

**Fig. 1 | Observed streamflow trends and their drivers in Brazil (1980–2015). a** Change in annual minimum 7-day streamflow (drought flows). **b** Change in annual maximum daily streamflow (flood flows). Blue and red indicate increasing and decreasing streamflow respectively (in % change relative to the long-term drought or flood flow, per decade). **c, d** Contributions to streamflow change in terms of

coefficients of two panel regressions between streamflow (n = 25,682 for droughts and 27,299 for floods) and mean daily P − E (precipitation minus evaporation), annual minimum 90-day P − E, annual maximum 14-day P − E, and water use. A coefficient of 0.5 indicates that a 1% change in a particular driver leads on average to a 0.5% change in drought or flood flows. Error bars represent the standard error.

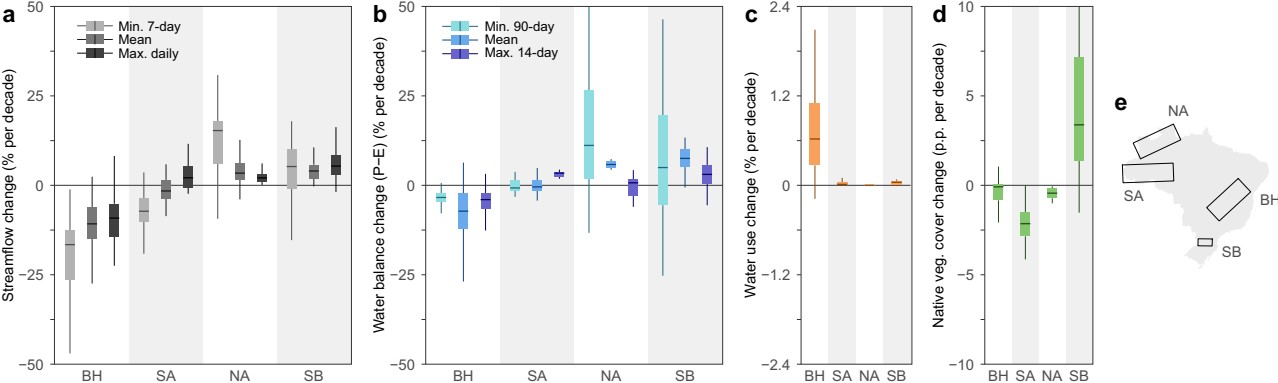

**Fig. 2 | Streamflow trends and contributing drivers in four hotspots of change.**
**a** Streamflow trends, with light grey boxes indicating minimum 7-day flows
(drought flows), medium grey indicating mean flows, and dark grey indicating
maximum daily flows (flood flows). **b** Climatic trends, with light blue boxes indi-
cating minimum 90-day precipitation minus evaporation (P − E), medium blue
indicating mean P − E, and dark blue indicating maximum 14-day P − E. **c** Water use
trends in % of the long-term mean daily streamflow per decade. **d** Native vegetation
cover trends in percentage points (p.p.) of the total area per decade. The boxplots
represent the spatial variability of the local trends within each hotspot. **e** The
hotspot locations (Northern Amazonia – NA, Southern Amazonia – SA, Southern
Brazil – SB, and Brazilian Highlands – BH). Boxplots show the median value, the first
and third quartiles, and 1.5 times the interquartile range. Outliers are not shown.

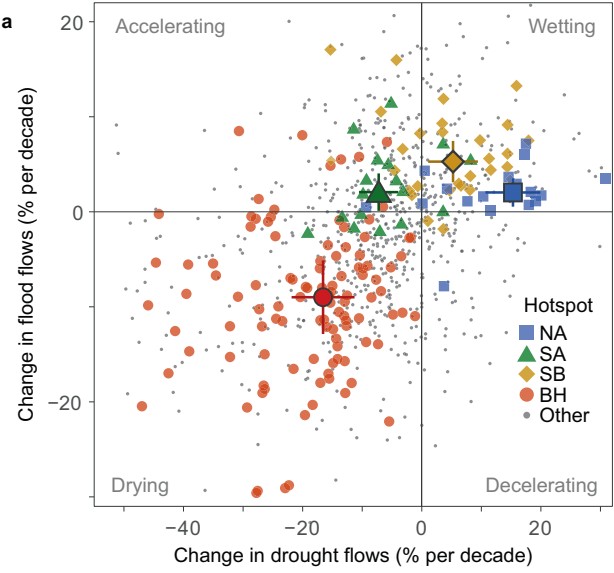

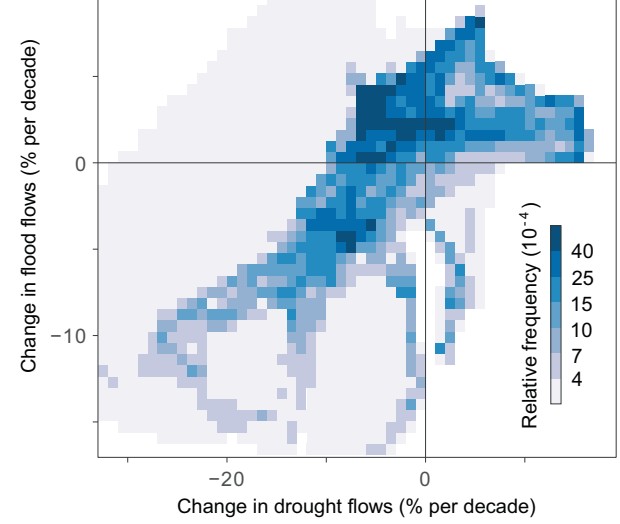

**Fig. 3 | Classification of streamflow trends into accelerating, decelerating,
wetting, and drying quadrants. a** Symbols without borders indicate flood and
drought flow trends of *n* = 886 stations. Hotspots (Northern Amazonia – NA,
Southern Amazonia – SA, Southern Brazil – SB, and Brazilian Highlands – BH) are
indicated by colors. Symbols with borders represent the median trend of each
hotspot, and the error bars indicate the median temporal uncertainty of the trend
estimates. **b** Classification of regional trends, with darker colors indicating higher
areal fraction per bin.

flood and drought flows are highly correlated (Spearman correlation in
the spatial variability of regional trends of 0.61, Fig. 3b), there is a
tendency towards an accelerating and drying water cycle. A total of
29% of the study area has been accelerating (Fig. 3b), which is double
the expected percentage of a standardized, bivariate normal dis-
tribution with a correlation of 0.61 (i.e., the correlation between
drought and flood flow trends; Eq. (5) in the Methods section). More-
over, 25% and 42% of the study area exhibit wetting and drying trends
respectively, whereas 35% would be expected in a standardized,
bivariate normal distribution. These figures are quite robust against
estimation uncertainty (Supplementary Fig. 9).

In order to analyze the causes of the acceleration of the terrestrial
component of the water cycle, we computed the average trend of each
driver from the locations associated with the bins of the bivariate
histogram of Fig. 3b (Fig. 4). Mean P − E trends are strongly positive (on
average + 5% per decade) and negative (on average −3% per decade) in

the wetting and drying quadrants, respectively (Fig. 4a), while they are
less important in the other quadrants. Increasing water use has
amplified the decreasing trends in the drying quadrant (Fig. 4c). Vir-
tually all areas where water use has increased by more than + 0.5% of
the long-term mean flow per decade are in the drying quadrant. The
accelerating quadrant is dominated by two factors. The first is
increasing trends in maximum P − E which has increased on average by
+ 2% per decade. The second factor is decreasing native vegetation
cover. A total of 60% of all areas where native vegetation cover has
decreased by more than 2 percentage points per decade are in the
accelerating quadrant.

## Discussion
While in the past some of the drivers of streamflow change such as
climate[28–35] and land management[36–42] have been analyzed individually
in South America, here we are showing a clear, spatially coherent signal

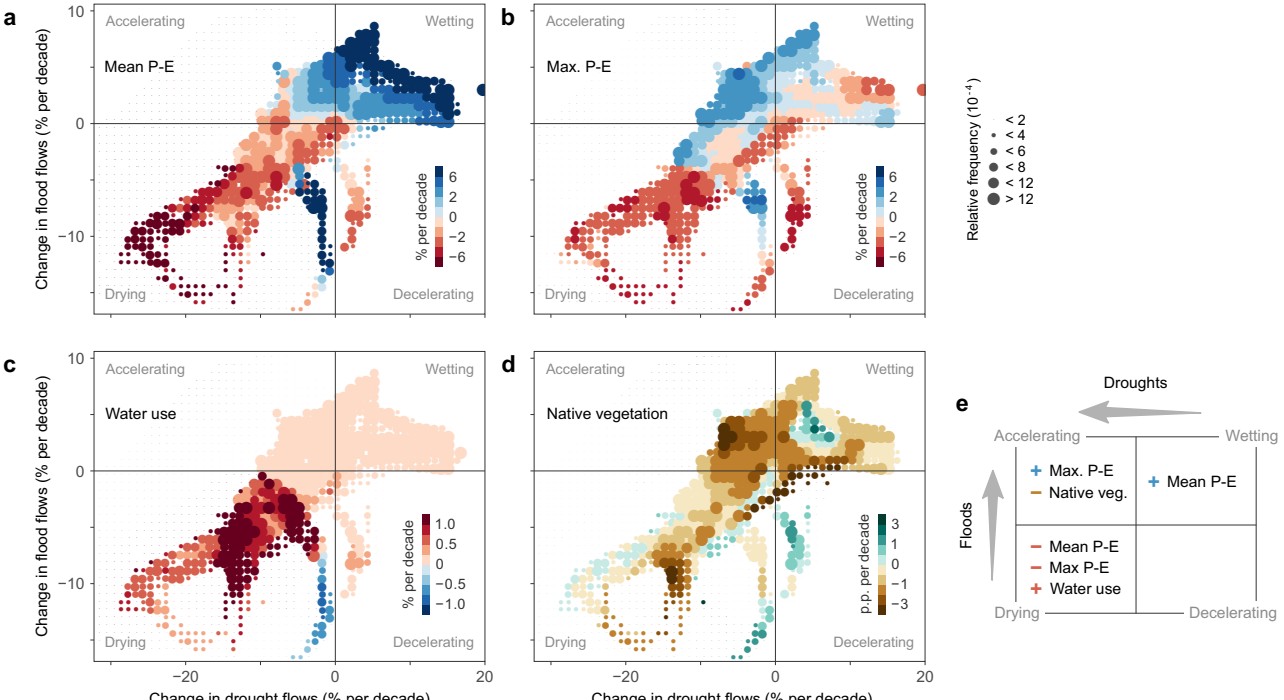

**Fig. 4 | Trends of the drivers mapped on the quadrants of the accelerating, decelerating, wetting, and drying streamflow trends of Fig. 3b. a** Mean daily precipitation minus evaporation (P − E). **b** Maximum annual 14-day P − E. **c** Water use in % of the long-term mean daily flow per decade. **d** Native vegetation cover in percentage points (p.p.) of the total area per decade. Larger circles indicate higher areal fractions of trends. Colors indicate the average trends of the drivers for each bin. **e** Schematic of the main drivers of streamflow changes.

of streamflow changes that can be interpreted in terms of the compound effects of these drivers. Drying trends are the largest in central and northeastern Brazil (Fig. 5). One possible explanation for the change is the southward shift of the South American Convergence Zone (SACZ), a major source of precipitation, which has moved away from central Brazil[29]. The drying trends may also be related to a northward displacement of the Intertropical Convergence Zone (ITCZ), which has moved the equatorial precipitation band farther away from northeastern Brazil[30]. Even though the average temperature has been increasing in central and northeastern Brazil over the past four decades[31], evaporation trends have been mostly not significant (Supplementary Fig. 5) possibly because of reduced precipitation water supply. An expansion of irrigated agriculture from 15 to 70 thousand km² (i.e., by 367%)[36] from 1980 to 2015 has led to a rapid growth of water abstraction, which in 2017 constituted 68% of the total Brazilian water use[37]. Increases in crop productivity and water demands due to a drier climate have boosted agricultural water use even further[38]. Water abstraction occurs mainly from May to September[36] during the dry season in most of central and eastern Brazil, which is linked to a substantial reduction in drought flows.

The northward shift of the ITCZ that has contributed to the reduced precipitation in northeastern Brazil has also contributed to the wetting trends in northern Amazonia[32,33], even though average temperature[31] and evaporation have increased. On the other hand, the wetting trends in southern Brazil might be associated with stronger effects of the El Niño-Southern Oscillation climate mode[34] and the strengthening and southwards shift of the SACZ[29].

An acceleration of the terrestrial water cycle has occurred extensively in southern Amazonia (Fig. 5). The northward shift of the ITCZ is linked to an expansion of dry season length[33] and warmer temperatures, which have increased evaporation particularly in southwestern Amazonia. Extreme wet-season precipitation and floods have increased as a result of the intensified ascending air masses of the

Walker circulation since the 1990s[28]. This intensification has been associated with warming trends of sea surface temperatures (SST) in the North Atlantic and cooling trends of SST in the tropical Pacific[28].

Another factor contributing to the acceleration trend of streamflow is deforestation, i.e., the substitution of tropical native vegetation by croplands and pasture, which has caused widespread land degradation in southern Amazonia[39,40]. Land degradation is associated with reduced soil infiltration capacity through soil compaction by agricultural machinery or grazing, reduction of soil fauna, and continued exposure of bare soil[23,43]. Consequently, surface runoff might increase and groundwater recharge decrease, thus both increasing floods and reducing the baseflow that maintains drought flows in the dry season[23]. This effect is particularly pronounced where streamflow is highly seasonal and drought flows depend on baseflow[44,45], as in southern Amazonia. Additionally, deforestation may increase extreme precipitation by triggering convection due to warmer land surface temperatures and increased patchiness[41]. On the other hand, deforestation may increase dry season length due to reduced moisture recycling[42] thus further extending hydrological droughts. Annual streamflow to rainfall ratios in three small southern Amazonian basins cultivated with soy beans was found to be twice that of neighboring forested basins, flows in the dry season were lower and those in the wet season were higher[46] similar to the present study. In contrast, analyses of streamflow in about 50 basins in Amazonia suggest that deforestation has increased low flows, likely because of decreasing transpiration, but without an effect on high flows[35,47], indicating that deforestation may potentially mask the effects of climate change on the water balance.

If the observed changes of extreme streamflow continue into the future in an unabated way, they will have substantial impacts in South America and on the global scale, some of which are already manifesting themselves. The impact will differ depending on where the region falls in the quadrant classification. In the Brazilian Highlands, for example, which lies in the drying quadrant, the drought flow of a

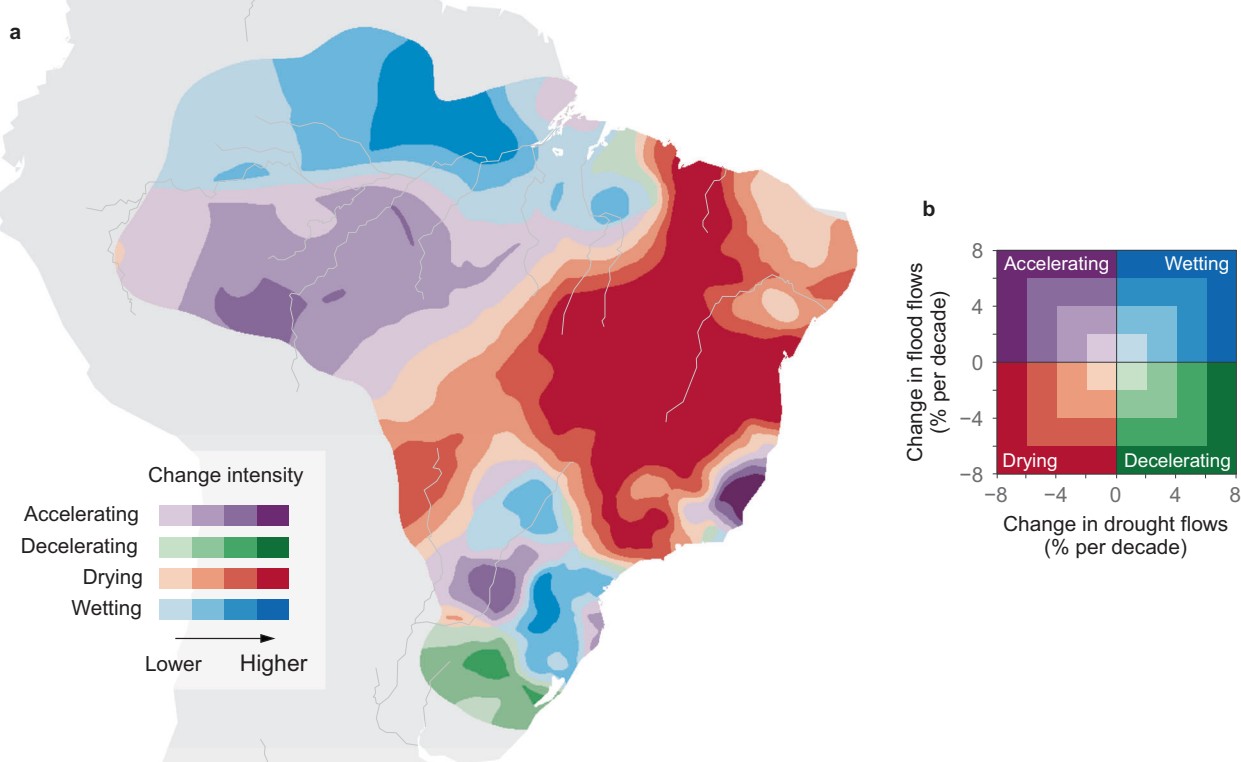

**Fig. 5 | Spatial distribution of the accelerating, decelerating, wetting, and drying streamflow trends in Brazil. a** The location of the four quadrants of regional streamflow trends, with darker colors indicating larger change intensities.

**b** Explanation of the color code of **a**. Accelerating water cycle has occurred in 29% of the region (2.7 million km²); deceleration in 4% (0.4 million km²); drying in 42% (3.9 million km²); and wetting in 25% (2.4 million km²).

10-year return period has become a 1-year drought (90% CI 1, 2) over the past four decades (Supplementary Fig. 10). Such an increase in drought risk threatens agricultural productivity and global food security[11]. During the 2012–2013 drought in Brazil and the U.S., global soybean prices soared to $550 per metric ton[48] and it is likely that these types of events will happen more frequently in the future. In wetting regions, flood hazards will increase and events such as the 2008 floods in São Paulo city, which caused damages on the order of 110 million USD[49], may become more frequent. In the urban areas of southeastern South America, those changes may exacerbate the coastal hazards resulting from sea level rise[50].

In regions with an accelerating water cycle, the situation of both droughts and floods will deteriorate if current changes continue. In Amazonia, where most of the hydropower potential is still untapped[51], future reservoir construction will have to account for the increased flood risk as the average 100-year flood in 1980 in the region has now become a 25-year flood (90% CI 5, 190) (Supplementary Fig. 10). Enhanced flooding may increase tree mortality through the inundation of floodplain forests[52], the "Achilles heel" of the Amazonian rainforest[53] and this process may be exacerbated by more intense droughts in the same region[12]. Reduced tree longevity could accelerate the transformation of Amazonia from a global carbon sink, currently sequestrating 0.4 Petagrams of carbon per year (25% of the terrestrial world total), to a global carbon source[12,13]. Reduced vegetation health can further reduce moisture recycling, increase the duration of dry spells and extreme precipitation events, potentially leading to a tipping point of forest dieback[42].

Given the evidence for the acceleration of the terrestrial water cycle demonstrated here for Brazil and by global climate model projections[1,54], similar climate and land management changes can also occur in other regions. It would therefore be advisable to conduct observation-based mapping studies globally. The evidence for the

acceleration found here also provides an opportunity for Earth System models to attribute the joint changes in floods and droughts to climate, deforestation and water use. In the face of still increasing carbon emissions and agricultural expansion, climate mitigation efforts need to go hand in hand with the adaptation of land management practices, in order to maintain food security and infrastructure safety through the compound risk management of floods and droughts.

## Methods
### Streamflow data
We used daily streamflow data from 886 hydrometric stations obtained from the CAMELS-BR dataset (Catchment Attributes and Meteorology for Large-sample Studies – Brazil)[55]. The hydrometric stations cover most of the largest river basins in tropical South America (Supplementary Fig. 1). The data have been collected following similar measurement protocols as the average of two daily staff gauge readings converted to streamflow through stage-discharge relationships. We selected the study period (1980 to 2015) as a trade-off between the number of stations and consistent record length since most hydrometric stations in the northern and western parts of Brazil were established in the second half of the 1980s. Only the stations that satisfied the following criteria were included in the study: (i) at least 25 years with less than 5% of data missing; (ii) with data starting before 1990; (iii) with data ending after 2005.

We removed hydrometric stations with typographic errors and unrealistically large discharges. Since we are interested in analyzing trends at a large scale, we also removed from the analysis the stations strongly affected by urban land cover (covering more than 10% of the basin area) or reservoirs (stations with a degree of regulation, i.e., the ratio of total reservoir storage capacity to total annual discharge, above 25%). The reservoirs considered were those from the CAMELS-BR data set, which is formed by a combination of data from the Global

Reservoir and Dam database (GRanD)[56], the Brazilian National Water Agency (ANA - Agência Nacional de Águas) Dam Safety Report 2017[57] and water bodies identified from Landsat satellite images[58]. The catchment boundaries were derived from the Global Streamflow Indices and Metadata Archive (GSIM)[59,60]. Catchment areas range from 11 km² to 5,120,000 km² with a median of 2080 km².

## Climate and land management data

We used daily precipitation time series from CHIRPS v2.0 (Climate Hazards Group InfraRed Precipitation with Station)[61] from 1981 to 2015. CHIRPS has a spatial resolution of 0.05° and includes data from rain gauges and satellite sensors. Our choice of using CHIRPS data was based on a comparison of precipitation trends from several gridded datasets (CHIRPS, MSWEP[62], PERSIANN[63], and CPC) with precipitation trends of ANA weather stations[64]. For the comparison, we estimated trends in the ANA dataset using daily precipitation data from 2315 weather stations with at least 25 years without missing values between 1980 and 2015. We interpolated the trends from the weather stations with ordinary kriging and correlated the interpolation with the trends of the gridded datasets. The trends of the CHIRPS and MSWEP data had the highest correlations with those of the weather stations (Supplementary Table 1a) and the median trends of the CHIRPS data were closest to those of the weather stations (Supplementary Table 1b).

Daily evaporation time series (including transpiration from vegetation) were obtained from GLEAM v3.3a (Global Land Evaporation Amsterdam Model)[65,66] from 1980 to 2015. GLEAM has a spatial resolution of 0.05° and is based on satellite soil moisture data and multiple meteorological products. We also conducted an alternative analysis using evaporation data from ERA5-Land[67,68] and obtained similar results.

We used global land cover data from ESA/CCI Land Cover v2.0.7 (European Space Agency/Climate Change Initiative) from 1992 to 2015 with a 300-meter spatial resolution and annual temporal resolution. We merged the following land cover classes: forests, shrublands, grasslands, sparse vegetation, and wetland. For simplicity, we denote this merged class as native vegetation cover.

Consumptive water use (i.e., abstracted water that does not directly return to the river basin) was extracted from the ANA's Manual of Consumptive Water Use in Brazil[37]. The data set is composed of monthly water use estimates for each municipality in Brazil from 1931 to 2015, classified into six categories: (i) irrigation, mapped from satellite images and characterized using national agricultural censuses; (ii) livestock, mapped from national agricultural censuses; (iii) households, estimated from the number of people in each municipality; (iv) industry, estimated from the number of employees in each industrial category; (v) mining, estimated from annual production; and (vi) cooling water for thermal power plants, estimated from annual production. Evaporation from reservoirs is not included in ANA's estimates, thus water use might be underestimated in some regions. We assumed water use in each municipality to be spatially homogeneous and converted the data to a 500-meter grid. Water use outside Brazil was not considered since the data were not available, but the main basins outside Brazil are in western Amazonia which has minor anthropogenic interventions.

## Trend analysis

For each hydrometric station we computed annual time series from 1980 to 2015 for the following variables: (i) minimum 7-day streamflow (drought flows), as it is widely used in Brazilian water management and trend analysis worldwide[69–74]; (ii) mean daily streamflow (water availability); (iii) maximum daily streamflow (flood flows); (iv) minimum 90-day precipitation (P) minus evaporation (E, including transpiration from vegetation); (v) mean daily P − E; (vi) maximum 14-day P − E; (vii) native vegetation cover; (viii) consumptive water use. The meteorological, native vegetation and water use variables are computed considering the contributing basin area of their respective hydrometric stations. The annual time series are computed in units of mm d⁻¹ so that their values are independent of basin size, except for native vegetation, which is computed in % of the basin area. Changing the time scales of maximum and minimum P − E did not modify the conclusions; the minimum 90-day P − E had correlations of at least 0.79 with other time scales ranging from 60 to 120 days; and the maximum 14-day P − E had correlations of at least 0.77 with other time scales ranging from 7 to 30 days. Similarly, changing the minimum 7-day streamflow and maximum daily streamflow by the 5th and 95th flow percentiles yielded similar results, as the Spearman correlations between their local trends are 0.94 and 0.72 respectively. In order to capture both the dry and wet seasons well, we used the water year from March to February for the minimum 7-day flow and minimum 90-day P − E; and from September to August for the other variables.

A linear trend magnitude in each annual time series (i.e., local trend) was estimated with the Theil-Sen slope estimator[75,76] (Supplementary Figs. 2–5). We evaluated the statistical significance of the trends with the Mann-Kendall test[77]. We removed significant (at the 5% level) lag-1 autocorrelation by trend-free pre-whitening[78]. We multiplied the trend magnitude by 10 to express it in terms of change per decade. The estimated local trend in each series was divided by the long-term average value of its own time series to transform it into units of % change per decade. For example, the lower Madeira river in southern Amazonia (gauge ID 15700000, latitude −5.8167, longitude −61.3019) has a drought flow trend of −0.00588 mm d⁻¹ yr⁻¹ and a long-term average drought flow of 0.4398 mm d⁻¹, which results in a trend of −13.4 % per decade. There are two exceptions to this transformation: (i) native vegetation cover, for which no transformation was necessary because the data is already in % of the basin area, therefore its trends are expressed in percentage points per decade; and (ii) water use, which was instead divided by the long-term mean daily streamflow because it is a more relevant index to relate to water abstractions.

We estimated regional trends (Fig. 1, Supplementary Figs. 2–5) by spatially interpolating local trends with ordinary block kriging using the gstat R package[79,80] and the best fit variogram models. The regional trends of the drivers are estimated by interpolating the local trends (which considers the contributing basin area of the hydrometric stations) so that it is consistent with the regional streamflow trends. The blocks are sized 4° by 4° (approximately 445 by 445 km at the equator), which allows for a robust analysis particularly in the Amazon (where gauge density is the lowest) with on average three gauges in each block. Interpolations using block sizes ranging from 1° by 1° to 6° by 6° yield similar results. The uncertainties of the estimated local trends (Fig. 3a) were evaluated with bootstrapping[81] (α = 0.34) and the uncertainties of regional trends with kriging standard deviations (i.e., kriging errors) (Supplementary Figs. 2–4). We checked the correlations between trends and catchment area to potentially account for its effect in the interpolation[82], but the Spearman correlations were close to null.

To evaluate the possible inflated variance effects on the spatial interpolation due to the spatial correlation between stations, which could lead to an overestimation of regional trends, we repeated the trend interpolation using two subsets of randomly selected stations: (i) using only stations with distances larger than 0.5° from each other; and (ii) using only stations with distances larger than 1° from each other (Supplementary Fig. 6). The spatial patterns of the trends are similar to those using all stations, with Spearman correlations of at least 0.93 between them. Additionally, we computed the regional Mann-Kendall test[83] for the trends in each hotspot of change. Changes in flood flows and drought flows are statistically significant for every hotspot (P < 0.001).

## Trend attribution

We evaluated the potential causes of streamflow trends with panel regressions (Fig. 1c, d), similarly to previous studies[84–86]. Panel regression includes time series data across multiple cross-sections (i.e.,

basins) in a single regression framework, allowing us to investigate the hydrological variability both in space and in time. We use fixed-effects (for location) regressions as we are mostly interested in analyzing the impacts of variables over time and as indicated by a significant ($P < 0.001$) Hausman specification test[87]. We compute two panel regressions, one for drought flows and another for flood flows. The drought flows regression has the form:

$$\ln(Q\min_{i,t}) = \beta_1 \ln(Pm_{i,t}) + \beta_2 \ln(P\min_{i,t}) + \beta_3 \ln(U_{i,t}) + \beta_4 V_{i,t} + \alpha_i + u_{i,t} \quad (1)$$

where $Q\min_{i,t}$ is the drought flow (i.e., minimum annual 7-day minimum flow) of basin $i$ at year $t$; $Pm_{i,t}$ is the mean daily P − E for that basin and year; $P\min_{i,t}$ is the minimum annual 90-day P − E; $U_{i,t}$ is the mean daily consumptive water use; $V_{i,t}$ is the percentage of native vegetation cover; $\beta_1$ to $\beta_4$ are the coefficients of the independent variables; and the last two terms are the error components, with $\alpha_i$ representing the intercept for basin $i$ and $u_{i,t}$ representing the idiosyncratic error. The flood flows regression has the form:

$$\ln(Q\max_{i,t}) = \beta_1 \ln(Pm_{i,t}) + \beta_2 \ln(P\max_{i,t}) + \beta_3 \ln(U_{i,t}) + \beta_4 V_{i,t} + \alpha_i + u_{i,t} \quad (2)$$

where $Q\max_{i,t}$ is the flood flow (i.e., maximum annual daily flow) of basin $i$ at year $t$; and $P\max_{i,t}$ is the maximum annual 14-day P − E. We use logarithms of mm d$^{-1}$ units for all variables except native vegetation as it is already expressed in percent coverage. Therefore, the regression coefficients can be interpreted in relative terms. For example, a 1% change in maximum annual P − E would lead to a $\beta_2$% change in flood flows assuming that the remaining independent variables are unchanged. We computed the standardized errors of the regression coefficients with robust covariance matrix estimators[88]. The regression analysis was performed with the R packages plm[89], sandwich[90,91] and lmtest.

The panel regressions were computed in two steps. First, we computed the regressions of Eqs. (1) and (2) for the years 1992 to 2015, which is the period covered by vegetation data. Both regressions had null and non-significant ($P > 0.01$) native vegetation coefficients. Thus, we removed the native vegetation terms and computed the regressions a second time including data from 1980 to 2015 (Fig. 1c, d). The regressions are robust to changes in the analysis period, with similar coefficients for the two time intervals analyzed (1992-2015 and 1980-2015).

We investigate the interannual variability of streamflow and its drivers in four hotspots with mounting environmental concerns (Fig. 2). The selected hotspots are located in the upstream areas of major South American basins with distinct streamflow regimes, land and water management. The Brazilian Highlands hotspot has widespread water-intensive crops with increasing drought and water scarcity issues[92,93], which covers the most arid regions upstream of the São Francisco and Paraná basins. The Southern Amazonia and Northern Amazonia hotspots have been under large-scale deforestation with potential hydrometeorological impacts[41,94,95], particularly in the south where land cover change is the highest[39,96]. The Southern Brazil hotspot has been under increasing flooding in recent decades[69,97], which covers the upstream areas of the subtropical Uruguay and Iguaçu basins. We note that the results are robust to variations in hotspot sizes (by ± 20%) and orientations (by ± 20°).

Following the methodology of a previous study[98], for each hotspot we standardized the annual time series at the stations of each variable to zero mean and unit variance to make the time series comparable within hotspots (Supplementary Fig. 7), for example:

$$Q_{i,k}^0 = \frac{Q_{i,k} - \mu_{Q_k}}{\sigma_{Q_k}} \quad (3)$$

where $\mu_{Q_k}$ is the mean and $\sigma_{Q_k}$ is the standard deviation of the streamflow time series for station $k$, from which we estimated the long-term mean $\mu_{Qh}$ for each hotspot and the square root $\sigma_{Qh}$ of the mean temporal variance. We compared the results between hotspots by denormalizing the series $k$ of each hotspot $h$:

$$Q_{i,k}^* = \sigma_{Qh} Q_{i,k}^0 + \mu_{Qh} \quad (4)$$

The hotspot time series (Supplementary Fig. 7) were smoothed using the LOESS method with a smoothing parameter of 0.5.

**Quadrant classification**
We examine how both flow extremes have changed in a single analysis using the quadrant classification. We classified the trends into four quadrants (Fig. 3): (i) wetting, when trends in drought flows and flood flows were positive; (ii) drying, when trends in drought flows and flood flows were negative; (iii) accelerating water cycle, when trends in drought flows were negative but those in flood flows positive; (iv) decelerating water cycle, when trends in drought flows were positive but those in flood flows negative. To analyze the role of the drivers in these changes we first computed a 2-dimensional histogram of regional trends in drought flows and flood flows (Fig. 3b). Then, we identified the spatial coordinates included in each 2-dimensional bin of the histogram. For each bin, we computed the average regional trends of the drivers at the associated spatial coordinates and plotted them along with their relative frequencies (Fig. 4 and Supplementary Fig. 8).

To determine the expected trend frequency in each quadrant, we considered the standardized, bivariate-normally distributed variables $\mathbf{Z_1}$ and $\mathbf{Z_2}$ evaluated with respect to regional trends in drought and flood flows. The quadrant probability can be evaluated[99] as

$$P(\mathbf{Z_1} \leq 0, \mathbf{Z_2} \leq 0) = P(\mathbf{Z_1} \geq 0, \mathbf{Z_2} \geq 0) = \frac{1}{4} + \frac{\sin^{-1}(\rho)}{2\pi} \quad (5)$$

where $\rho$ is the correlation of $\mathbf{Z_1}$ and $\mathbf{Z_2}$. A significant correlation between changes in drought and flood flows is expected as they are often consistent with each other and the entire flow distribution moves either upward or downward[5,6]. Here, we set $\rho$ to 0.61, corresponding to the spatial correlation between the regional trends in drought and flood flows found in the trend analysis in the present study (Fig. 3b). According to Eq. (5), in a random set, 15% of the trends would be expected to fall in each of the accelerating and decelerating quadrants and 35% in each of the drying and wetting quadrants.

To demonstrate the robustness of the results, we examined the sensitivity of trend frequency in each quadrant as a function of the spatial uncertainty in regional trends (i.e., the kriging errors) (Supplementary Fig. 9). Even if 30% of the locations with the highest average kriging errors were not considered in the analysis, the areal coverage of the accelerating quadrant would only change from 29% to 24%, which is still well above that of a random sample (i.e., 15%). The drying quadrant becomes even more frequent and the wetting quadrant less frequent as the locations with the highest kriging errors are not considered.

**Changes in the return period**
In evaluating observed changes in the return periods of drought and flood flows (Supplementary Fig. 10), we followed a previous study[98] where the location parameter of the probability distribution is allowed to change with time. For compatibility of streamflows in catchments of different sizes, this analysis was made using streamflow per unit catchment area. The probability density function $f(x)$ of the annual

maximum, $x$, was estimated for each station using a generalized extreme value distribution (GEV)

$$f\left(x\,|\,\mu_t,\sigma,\xi\right)=\frac{1}{\sigma}\left[1+\xi\left(\frac{x-\mu_t}{\sigma}\right)\right]^{-\left(\frac{1}{\xi}+1\right)}\exp\left\{-\left[1+\xi\left(\frac{x-\mu_t}{\sigma}\right)\right]^{-\frac{1}{\xi}}\right\}\quad(6)$$

where $\mu$ is the location, $\sigma$ is the scale, and $\xi$ is the shape parameter of the GEV distribution. The location parameter, $\mu_t$, changes linearly with time $t$ as

$$\mu_t=a+bt\qquad(7)$$

The parameters $a$, $b$, $\sigma$, and $\xi$ were estimated from the maximum flow series using Bayesian inference through a Markov chain Monte Carlo (MCMC) with the Differential Evolution Adaptive Metropolis (DREAM$_{(ZS)}$)[100,101]. Non-informative uniform prior distributions were used for $a$, $b$, and $\sigma$, whereas a normal distribution consistent with the geophysical prior[102] was used for $\xi$. We drew 12,000 parameter samples from the posterior distributions, from which 12,000 100-year flood flows in 1980 were calculated for each station by inverting the cumulative distribution function of the GEV and using Eq. (7) with $t$ = 1980. The changed return period of these 12,000 flood flows in 2015 were computed using the cumulative distribution function of the GEV and using Eq. (7) with $t$ = 2015. Finally, the median of the 12,000 return periods was used as the 2015 return period of the 100-year flood flow in 1980.

In the case of low flows, Eq. (6) was used after taking the negative of the original minimum 7-day flow series. The parameters $a$, $b$, $\sigma$, and $\xi$ were estimated using the same MCMC algorithm[100,101] with non-informative priors for all parameters in this case. We drew 12,000 parameter samples from the posterior distributions, from which 12,000 10-year minimum 7-day flows in 1980 were calculated. The changed return period of these 12,000 drought flows in 2015 were computed using the cumulative distribution function of the GEV and using Eq. (7) with $t$ = 2015. Finally, the median of the 12,000 return periods was used as the 2015 return period of the 10-year drought flow in 1980. Those stations for which the 5th and the 95th percentiles of the uncertainty distribution agreed in the sign of change are plotted as large points in Supplementary Fig. 10, whereas the remaining stations are plotted as smaller points to indicate the uncertainty involved in the estimation.

## Data availability

Daily streamflow data are available at https://doi.org/10.5281/zenodo.3709337 and http://www.snirh.gov.br/hidroweb/. Daily precipitation data from CHIRPS v2.0 are available at https://www.chc.ucsb.edu/data/chirps. Daily precipitation data from MSWEP are available at https://www.gloh2o.org. Daily precipitation data from PERSIANN are available at https://doi.org/10.7289/V51V5BWQ. Daily precipitation data from CPC are available at https://psl.noaa.gov/data/gridded/data.cpc.globalprecip.html. Daily evaporation data from GLEAM v3.3a can be downloaded from https://www.gleam.eu/. Daily evaporation data from ERA5-Land can be downloaded from https://doi.org/10.24381/cds.e2161bac. Global land cover data from ESA/CCI v2.0.7 can be downloaded from http://maps.elie.ucl.ac.be/CCI/viewer/download.php (© ESA Climate Change Initiative - Land Cover led by UCLouvain, 2017).

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

## Acknowledgements

This work was supported by the Brazilian National Council for Scientific and Technological Development (CNPq) grant 141219/2019-0 (V.C.), grant 201343/2020-7 (P.C.), and grant 314792/2020-1 (P.C.) and the Austrian Science Funds (FWF) projects I 3174, I 4776 and W1219-N22 (G.B.). Debora Yumi de Oliveira provided code for the change in return period analysis.

## Author contributions

V.C., P.C., and G.B. designed the study, interpreted the results, and prepared the manuscript. V.C. processed the data, conducted the analyses, and created the figures. P.C. conducted the change in return period analysis.

## Competing interests

The authors declare no competing interests.
