## [Peer Review File · Nature Communications]

Climate and land management accelerate the Brazilian water cycleReviewers' Comments:

Reviewer #1:

Remarks to the Author:

Review of Chagas et al., (2021).

The study of Chagas et al., (2021) attempts to explain the spatial patterns of changes in the hydrologic regimes of Brazilian watersheds with respect to the relevant spatial patterns in the local climate, and land management (=water use and land-use change). The authors used a dataset containing streamflow measurements for Brazilian catchments (the CAMELS-BR dataset, presented in Chagas et al., (2020)) to obtain the trends in hydrologic behavior (i.e., streamflow-based metrics representing mean streamflow, floods, and droughts) over the period of 1980-2015. They have spatially interpolated those streamflow-trends to be able to proceed with the inference of which (spatially distributed) drivers appear to be responsible for such changes. The results explore the interactions between climate and human intervention land-management as drivers of the observed changes.

I think the paper is well written and the statistical analysis is exhaustive and well documented (with just a few points deserving clarification). I also think the results are of great importance, given Brazil's dimension and importance to the global economy and climate.

I have one main concern. The paper is presented as a South America-based analysis. However, all data used, and its spatial distribution is constrained to the Brazilian territory, and not the South American continent. The local factors determining how land-management and climate interact give rise to the observed changes in hydrologic behavior are representative of the intrinsic historical and geopolitical decision-making processes within Brazil. This should be evident to any reader, but no effort was made by the authors to reconcile this, which makes the text misleading in many parts.

I think the readers of this study, especially the ones from other countries in South America will ask the same questions I am posing here. These are questions that are, in my opinion, very important if the study has indeed the goal of reporting changes in the water cycle in South America. (i) What are the hydroclimatic patterns taking place across the continent (streamflow trends, P and E trends)? (ii) What are the patterns of human intervention (water use and land use change) throughout South America? (iii) Are there other hotspots of change occurring outside Brazil?

If the real intent is to provide an analysis of South America, the authors should expand their analysis to accommodate the above questions. Otherwise, the authors should clearly indicate the fact that this study investigates the Brazilian territory and no other regions within South America. This should be done all through the manuscript, with especial attention to the title and abstract, which are not providing a clear description of what has been done in the study. Therefore, I recommend major revisions.

Minor Comments:

L19: 42% of South America is comprised by agricultural zones? 42% percent of South America is experiencing drying? The parenthesis is causing a bit of confusion here.

L34-37: I don't quite follow the causality between atmospheric moisture "carrying" capacity with enhanced extreme rainfall, potential evapotranspiration, and rainfall seasonality.

- Regarding potential evapotranspiration: if the inclusion of CO₂ effects on stomatal closure (therefore surface resistance) are included, E_p might not necessarily increase:

Yang, Y., Roderick, M.L., Zhang, S. et al. Hydrologic implications of vegetation response to elevated CO₂ in climate projections. *Nature Clim Change* 9, 44–48 (2019). <https://doi.org/10.1038/s41558-018-0361-0>

Milly, P., Dunne, K. Potential evapotranspiration and continental drying. *Nature Clim Change* 6, 946–949 (2016). <https://doi.org/10.1038/nclimate3046>

- It might be a bit of stretch to link increasing moisture carrying capacity to seasonality, directly. Changes in seasonality occur as a consequence of large scale atmospheric circulation.

L37-38: How do these last 3 factors increase flood magnitude and exacerbate low flow conditions? Are you saying that in reference to rainfall seasonality only? I think this sentence could be clearer.

L52-54: The dataset used in this study contains predominantly data for Brazil, not the whole South American continent. Therefore, this study is primarily an investigation of how climate change might be affecting Brazilian water resources.

I think the authors should make an effort to explicitly address this and change the language throughout the whole text, especially the title. Additionally, the main feature of this study is to bring to light the link how human intervention (captured as the non-climatic drivers of water consumption and reduction in forest cover) has been interacting with climatic drivers to alter the hydrologic cycle. Therefore, one cannot argue that South America is undergoing the same changes, unless it is clear that all other countries experience the same behavior with respect to its non-climatic drivers.

L59: How was a timeseries of mean annual streamflow produced over only 25 years data? If you're dealing with annual data, what you might have is mean daily (or monthly) discharge over each year.

L64: I am also a bit confused here, regarding (i): what is the sample length here and what was the number of years assumed?

L68: Not sure if I follow the assumption of 14 days and its relationship between large and small basins.

L74: Brazil, not South America

Figure1:

Figure1, insets. Shouldn't the contributions add up to 1? What are the units on the y axis?

L99-100: I don't understand the choice of "mean annual per decade". From what I understood from reading the methods, the trends were calculated based on annual data. Therefore, why aren't the values reported as "per year"? Based on how many data points were the regressions calculated?

L105: What is the rationale behind the choice of the hotspots? Why were they picked? This is not clear in the text. Do the results arising from hotspot analysis vary with the choice of different rectangles? In other words, are these hotspots decided based on an intrinsic characteristic? Please try to explain better why the rectangles were placed there and not in other places.

L106: sign, instead of signal?

L107: Maybe once you define, it would be good to use acronyms for the hotspot names such as AS, AN, SB, BH. Maybe a number/letter system? Also, since the hotspot analysis is an explicit part of the results, I recommend including the hotspot bounding boxes in Figure 1 already (with acronyms, or number system, etc.). I see after reading the extended data that acronyms and number system were used. Make sure to follow one method only.

Figure3: Only here you defined acronyms for the hotspots. It will read better if you introduce them

L194: Please instruct the reader in the regional climates in Brazil. Is the dry region from May through September a constant across the whole country?

L227-228: Reference needed here.

L445: Not sure if I follow the definition of mean annual here. Mean annual should refer to the average of many annual data. How did you calculate the mean annual data per year? I think what was actually done was a mean daily flow per year, or the total annual flow. Please provide a clear description of how that was calculated.

L465-467: What is the reason for the choice of 4x4 degree? How sensitive are the results with respect to block dimensions?

L494-496: If panel regression is used to explain the trends, wouldn't it make more sense to have performed trend analysis and panel regression for the same time interval? Does the trend analysis of streamflow-based metrics vary if the 1980-1991 period is excluded?

Reviewer #2:

Remarks to the Author:

Summary:

I find this paper to be an interesting and valuable contribution to the existing body of literature concerning the compound land and climate drivers of water balance change in Brazil. While the paper is an interesting application of data and methods, some of the methods descriptions are so unclear as to inhibit understanding of the study and interpretation of the results. At present, the findings -- while interesting -- do not appear particularly novel, given the literature documenting similar dynamics in this region. I make specific suggestions below in hopes of helping the authors revise and improve their manuscript.

Major comments:

Abstract: The abstract suggests that this study demonstrates cause and effect: "water use and climate change have amplified..."; "Drying... is due to..."; "Acceleration... is linked to...". The analysis, insofar as I understand it, shows potential associations in time and space between climate and land cover, but does not establish clear causal connections. Therefore, this language should be revised, unless the methodological approaches used in the study can be clarified (see comments below) and shown to support a causal interpretation.

l.58-60 There is no explanation (in the main text or methods) for the selection of the drought (min 7-day) and flood (max daily) flow statistics. Why was the minimum pulled for 7-day periods, but the maximum from daily? Additionally, because these statistics give only one observation per year, the local time series trends (which are then interpolated to the region, and used for subsequent analyses) appear to be calculated based on only 35 observations (1980-2015). I understand there's not much to be done about the limited time range, but it does make me wonder why a more 'stable' statistic wasn't used for the min/max streamflow statistics, like the 5th and 95th, or 10th and 90th percentiles of 7-day or daily streamflow instead. Those percentiles would still be relevant to 'drought' and 'flood flow' conditions, just not the most extreme drought and flood flows. An analysis of true extremes might be justifiable by using the current min/max statistics aggregated over all sites (across sites), but the within-site design of this study (interpolated local trend analysis), wherein trends are fundamentally based on a sample size of 35, doesn't seem well suited to the use of the most extreme observations.

l.70-71 I appreciate the concision of this sentence, but in order to understand the analysis and results as presented in the remaining text, it's necessary for the methods to be explained in more depth. I had difficulty understanding the results because I didn't sufficiently understand the study design; I was frequently jumping back and forth between the main text and methods. The analysis and results should be generally understandable from the main text alone, and that is not presently the case. Specifically, the approach used for each component of the analysis (flow trend detection, attribution of streamflow trends to drivers, focus on hotspots, quadrant analysis, return flow analysis) needs to be summarized in the main text -- either at this location in the text (introduction), or just prior to the discussion of each result. The panel regression model needs to be written out in the Methods at a minimum. At present, there is no model description/equation: I don't know the structure of the model, I don't know how the fixed effects were set up, and therefore I cannot assess to what extent the regression analysis functions in the way it is intended.

l.77-onward. The units (% per decade) in drought and flood flows are not clearly defined, and difficult to interpret. For example, the sentence "Regional trends in drought flows range from -37 to +16% per decade..." is unclear in terms of what a trend of "% per decade" actually means. I interpreted this to mean that there was a reduction the size of 37% of some baseline flow in some locations, up to an increase the size of 16% of some baseline flow in other locations, over a 10 year period. In the Methods (l.459) I see that these baseline flows are the average annual flows, and that dividing by the average annual flows was done to "transform it [flows] into units of % change per year". From what I can tell, that is not what the division achieves. Dividing annual drought or flood flow change/year by average annual flow makes the units: 'flow change/year expressed a % of mean annual flow', not "%

change per year". I read those as different things. For example, if you saw an annual change in flood flows in some basin equal to 10 m³/year, and the average annual flows in that basin are 100 m³, that annual change of 10 m³/year is equal to 10% of average annual flows -- it's not a 10% change per year. This also creates confusion for interpretation of decadal changes: are the decadal changes just the sum of annual trends: X m³/year (expressed as % of average annual flow) * 10 years? Or, are the decadal changes an actual "percent change" between years: (flow_year 10 - flow_year 1) / flow_year 1 * 100? The same lack of clarity stands for the other variables as well (P, E, water use, forest cover). With respect to water use, it looks like trends there would be change expressed as a percent of *long term* average annual flow (instead of average annual flow). If forest cover is reported in units of % of basin area, then a change in forest cover from year to year, or over a decade, is actually a "percentage point" change -- a change between two percentages. The specifics of this are really important for interpretation, and at present these specifics are unclear in the results text, in the figures, and in the methods.

Lastly, for the trend analysis, I was unable to locate a location in the paper where the units of streamflow (volumetric, or basin area-averaged depths) are indicated; the regression analysis (l.491) and return flow analysis (l.556-557) discussions in the Methods note units of depth, but it's unclear what's used for the other analysis components. For the return period analysis, you write the use of depths is to address "compatibility of streamflows in catchments of different sizes." There is a need for this same compatibility in the trend and other analyses as well, but it's unclear what streamflow units were used. For example, smaller basins may show more variable flow, potentially creating instability in their drought and flood flow trends relative to larger basins.

l.99 Are the percent changes here the same as those reported in the main text and in the methods? "percent change of the mean annual value per decade" could mean several things... see comment above.

l.100 the panel regression setup, and the interpretation of panel regression coefficients (including units) needs to be explained in the main text and methods before the figure insets can be understood.

l.105 the delineation of the four hotspot regions should be defined more clearly, at a minimum in the Methods. It would help to include a small map of the hotspot locations as well in the main text (along w/ Fig 2), as the hotspot locations only appear in the Extended Data figures. It is not clear how the precise extent and location of these hotspots was determined. I understand the general regional placement, but why the chosen sizes and orientations? In what way was climate, water use, forest cover, and "floods and drought processes" used to determine the locations?

l.107 - 117: Because the analysis has not yet addressed formal attribution (Fig 2 is just a visualization of univariate distributions), consider revising this part to make it clear these are observations of associations: saying "reduction of drought flows is related to decreasing mean P-E" isn't really shown by Fig.2 -- it just looks like they occur at the same time.

l.107-118: This is an interesting discussion, in part because it is at odds to some extent with studies that show deforestation has different and complex effects in this region. Prior literature suggests that removing forest may increase baseflow or mean flow, while not affecting flood flow, with varying interpretations of the role of simultaneous climate variability and change -- deforestation has been shown to potentially mask the effects of climate change on the water balance. Referencing that literature, and discussion of this and other findings in your paper relevant to that literature, is warranted. See:

<https://doi.org/10.5194/hess-21-1455-2017>

<https://doi.org/10.1002/2017GL076526>

<https://doi.org/10.1111/j.1365-2486.2011.02392.x>

<https://doi.org/10.1007/s10584-020-02736-z>

<https://doi.org/10.1016/j.ejrh.2020.100755>

I.135-137: What estimate of correlation is this referring to: is this temporal (time series) or spatial correlation?

I.138-140: This finding is unclear because the Methods (for the quadrant analysis and other analyses) are insufficiently summarized in the main text. See comment above -- there needs to be a more detailed summary of methods for each analysis component in the main text so that the results can be understood.

I.177 - 180: this sentence implies that this paper is novel with respect to evaluating compound effects, which is not entirely true (see above comment and suggested papers), although this study is certainly an important contribution to that literature. Again, incorporation of discussion of previous studies of compound effects -- specifically -- here and throughout this section is warranted.

I.176 - 232 (section): the majority of this section reads like background information or a literature review, not interpretation/discussion of your results specifically. I'd suggest that most of this material be summarized more concisely and moved to the introduction, and then retain only aspects of this information that provide direct context for your specific results here.

I.239-240, 253-254: it's not possible to understand the return period results as they are written because the GEV approach/method was not introduced previously in the paper. Again, this method and its purpose, along with all other analysis approach methods, need to be introduced and summarized more clearly earlier in the main text, not just the Methods section at the end. Otherwise, it's not possible to understand the paper.

I.264-272: this final paragraph is very generic and vague, and adds little insight or value to the paper. Suggest deleting and/or replacing with recommendations that stem from specific findings of this study. For example, it's unclear (in part because of grammar -- see minor comment below) what the suggestion about use of ESMs means, and how it stems from your findings in particular.

I.388-389: The way it is written, I'm not able to tell if you are using data from CAMELS-BR or obtaining data from ANA, but formatting it similar to CAMELS-BR.

I.424-425: Given that all the listed non-forest land cover categories can have biological (water consumption) properties distinct from deep-rooted forests, particularly in this region, I'd suggest NOT using the term "forest" to describe this aggregation of land types; "natural", "native", or "uncultivated" lands might be more appropriate. Given the prevalence of savanna (shrubland and grassland) vegetation over large swaths of your study region, the term is important.

I.443: This suggests all P, E, Q statistics (for the trend analysis at least) were computed at the location of the streamflow stations/locations only -- that is, trends in P-E variables were computed at the point/station where streamflow was collected, NOT over the contributing basin area of that streamflow point/station. Is that correct (I'm assuming not, but the paper doesn't make it clear)? The appropriate comparison of streamflow and drivers would come from basin area-averaged estimates of drivers rather than point estimates. If P-E, water use, and forest were NOT computed over the contributing basin areas, then this should be done.

I.458-463: see comments above about confusion regarding the definition of % change in all variables.

I.474: I appreciate the check here, but don't entirely understand if this alleviates the concern, particularly for regression analysis. Can you explain what the concern is regarding inflated variance: what would this do to your trend (or other, i.e. regression) analyses?

I.486: it is necessary that the regression equations be written out in the text - at a minimum in the Methods, and ideally (in some concise form) in the main text. Stating that you used panel regression with fixed effects similar to other studies (only) is insufficient, and makes it impossible to review the quality of the regression component of this analysis. Specifically: what fixed effects (for time and/or location) were used?

I.498: similar results in terms of what? In terms of the coefficients on the independent variables?

I.498-500: this is insufficiently explained. What are the trends -- from the trend analysis? Per year trends, or the per-decade trends shown in the trend analysis results? What do you mean by similar? Similar in terms of the signs of the coefficients, the magnitude of the coefficients? This is confusing because coefficients from your main regression setup (assuming a simple log~log regression) would be: "a one percent change in flow/year is associated with a one percent change in driver/year"; for the trend regression, based on what I'm reading, the coefficient interpretation would be "a percent change in flow/year² is associated with a one percent change in driver/year²" (in the trend regression case, the coefficient pertains to 'a change in a change in flow', rather than 'a change in flow').

I.525 It's not clear if for this analysis you also used driver data summarized at the streamflow station locations, or all (interpolated, gridded) locations. In the case of drivers, did you use observations interpolated from the summaries of drivers made at the streamflow station points, or the original gridded data for P-E, water use, forest? The way the main text and methods are written, it's really unclear where you are using original gridded datasets, interpolations made from data summaries done at the streamflow station locations, or data summaries done at the streamflow station location (points). Are the background colors in Extended Data Figs. 3,4,5 from the original gridded data or the interpolations of point-level data summaries?

I.530-532 It would be helpful to describe this method, and its rationale, in a bit more detail so that readers understand the motivation behind its use and the value of the contribution. For example, it would help to just say Z1 and Z2 were evaluated with respect to drought and flood flows. When discussing rho being set by "regional trends in correlation" - what trends and what correlation: temporal or spatial correlation? From the trend analysis? What component of "this study" does this refer to? In general, I don't understand the intuition behind or motivation for the quadrant analysis approach because it's insufficiently explained.

Minor comments:

I.106 signal should be signals

I.235 - 262: check grammar throughout these paragraphs: need spaces between instances of "%" and "CI"; delete "have", and add "will" after "events" (I.243); delete "have", and replace "in" with "on" after "damages" (I.245).

I.267-268: this is vague.

I.267-269: check grammar. "including land management effects" is unclear - what does this mean?

I.467: why do you suggest this scale? Just explain.

Extended Data Fig. 7: why isn't forest included in panel (a)?

Reply to comments by Referee #1

We appreciate the comments of Referee #1 that helped us clarify the main points of the manuscript. Please, find below our replies to the comments.

The authors' response is printed in blue.

General comment:

(1) The study of Chagas et al., (2021) attempts to explain the spatial patterns of changes in the hydrologic regimes of Brazilian watersheds with respect the relevant spatial patterns in the local climate, and land management (=water use and land-use change). The authors used a dataset containing streamflow measurements for Brazilian catchments (the CAMELS-BR dataset, presented in Chagas et al., (2020)) to obtain the trends in hydrologic behavior (i.e., streamflow-based metrics representing mean streamflow, floods, and droughts) over the period of 1980-2015. They have spatially interpolated those streamflow-trends to be able to proceed with the inference of which (spatially distributed) drivers appear to be responsible for such changes. The results explore the interactions between climate and human intervention land-management as drivers of the observed changes.

I think the paper is well written and the statistical analysis is exhaustive and well documented (with just a few points deserving clarification). I also think the results are of great importance, given Brazil's dimension and importance to the global economy and climate.

Major comments:

(2) I have one main concern. The paper is presented as a South America-based analysis. However, all data used, and its spatial distribution is constrained to the Brazilian territory, and not the South American continent. The local factors determining how land-management and climate interact give rise to the observed changes in hydrologic behavior are representative of the intrinsic historical and geopolitical decision-making processes within Brazil. This should be evident to any reader, but no effort was made by the authors to reconcile this, which makes the text misleading in many parts.

I think the readers of this study, especially the ones from other countries in South America will ask the same questions I am posing here. These are questions that are, in my opinion, very important if the study has indeed the goal of reporting changes in the water cycle in South America. (i) What are the hydroclimatic patterns taking place across the continent (streamflow trends, P and E trends)? (ii) What are the patterns of human intervention (water use and land use change) throughout South America? (iii) Are there other hotspots of change occurring outside Brazil?

If the real intent is to provide an analysis of South America, the authors should expand their analysis to accommodate the above questions. Otherwise, the authors should clearly indicate the fact that this study investigates the Brazilian territory and no other regions within South America. This should be done all through the manuscript, with especial attention to the title and abstract, which are not providing a clear description of what has been done in the study.

Reply: We have changed the study area from "South America" to "Brazil" throughout the manuscript so that it is more precise. A study including the whole of South America would be impractical because large parts of the continent have limited data for streamflow, water

abstraction, and water regulation for the period analyzed (1980-2015).

We have, however, referred to the study area in the abstract as “major South American tropical river basins” (line 16 of the revised manuscript). In fact, out of the six largest river basins in South America, all but one (the Orinoco) are included in the analysis. And, even though most streamflow gauges are located inside Brazil, the associated changes integrate the climate and land management over the entire river basins, a significant portion of which falls outside Brazil’s territory (Extended Data Fig. 1).

Minor comments:

(3) L19: 42% of South America is comprised by agricultural zones? 42% percent of South America is experiencing drying? The parenthesis is causing a bit of confusion here.

Reply: We have clarified it: “Drying (fewer floods and more droughts) is aligned with decreasing rainfall and increasing water use in agricultural zones and occurs in 42% of the study area.” (lines 18-20 of the revised manuscript).

(4) L34-37: I don’t quite follow the causality between atmospheric moisture “carrying” capacity with enhanced extreme rainfall, potential evapotranspiration, and rainfall seasonality.

Reply: We have clarified the sentence and changed it to “In a warming climate, the moisture carrying capacity of the atmosphere is increased¹⁴ enhancing extreme rainfall^{15,16} which may increase streamflow during floods. Enhancement of rainfall seasonality¹⁷ may decrease streamflow during hydrological droughts.” (lines 35-38 of the revised manuscript).

(5) Regarding potential evapotranspiration: if the inclusion of CO2 effects on stomatal closure (therefore surface resistance) are included, Ep might not necessarily increase:

Yang, Y., Roderick, M.L., Zhang, S. et al. Hydrologic implications of vegetation response to elevated CO2 in climate projections. *Nature Clim Change* 9, 44–48 (2019).
<https://doi.org/10.1038/s41558-018-0361-0>

Milly, P., Dunne, K. Potential evapotranspiration and continental drying. *Nature Clim Change* 6, 946–949 (2016). <https://doi.org/10.1038/nclimate3046>

Reply: We have removed “potential evaporation” from the sentence (see Comment #4).

(6) It might be a bit of stretch to link increasing moisture carrying capacity to seasonality, directly. Changes in seasonality occur as a consequence of large scale atmospheric circulation.

Reply: We have rephrased the sentence for clarification (see Comment #4).

(7) L37-38: How does these last 3 factors can increase flood magnitude and exacerbate low flow conditions? Are you saying that in reference to rainfall seasonality only? I think this sentence could be clearer.

Reply: We have rephrased the sentence for clarification (see Comment #4).

(8) L52-54: The dataset used in this study contains predominantly data for Brazil, not the whole South American continent. Therefore, this study is primarily an investigation of how climate change might be affecting Brazilian water resources.

I think the authors should make an effort to explicitly address this and change the language throughout the whole text, especially the title. Additionally, the main feature of this study is to bring to light the link how human intervention (captured as the non-climatic drivers of water consumption and reduction in forest cover) has been interacting with climatic drivers to alter the hydrologic cycle. Therefore, one cannot argue that South America is undergoing the same changes, unless it is clear that all other countries experience the same behavior with respect to its non-climatic drivers.

Reply: We have changed the study area from “South America” to “Brazil” throughout the manuscript, including the title, for a more precise description (see Comment #2).

(9) L59: How was a timeseries of mean annual streamflow produced over only 25 years data? If you’re dealing with annual data, what you might have is mean daily (or monthly) discharge over each year.

Reply: We computed the mean daily discharge over each year from 1980 to 2015. We also changed from “mean annual streamflow” to “mean daily streamflow” throughout the manuscript. and revised the sentence to clarify it: “For each station, we compute annual time series of annual minimum 7-day streamflow as a measure of drought flows, mean daily streamflow as a measure of water availability, and annual maximum daily streamflow as a measure of flood flows.” (lines 57-60 of the revised manuscript).

(10) L64: I am also a bit confused here, regarding (i): what is the sample length here and what was the number of years assumed?

Reply: We have clarified the data sample length and changed from “mean annual atmospheric water balance” to “mean daily atmospheric water balance” throughout the manuscript so that it is consistent with the changes from Comment #9 (mean daily streamflow). The revised sentence is: “For each basin, we consider three climate drivers of streamflow change, computed from daily meteorological data from 1980 to 2015: (i) mean daily atmospheric water balance, computed as precipitation (P) minus evaporation (E, including transpiration from plants)” (lines 65-68 of the revised manuscript).

(11) L68: Not sure If I follow the assumption of 14 days and its relationship between large and small basins.

Reply: We have clarified and updated the sentence to “... (iii) annual maximum 14-day P – E because, as basin response times range from less than a day in small basins to a few months in large basins, the 14-day time scale is a compromise on which basins are most sensitive.” (lines 69-71 of the revised manuscript).

(12) L74: Brazil, not South America

Reply: We have corrected it (line 85 of the revised manuscript).

(13) Figure1: Figure1, insets. Shouldn't the contributions add up to 1? What are the units on the y axis?

Reply: We have now clarified in the Figure caption how to interpret the regression coefficients: "A coefficient of 0.5 indicates that a 1% change in a particular driver leads on average to a 0.5% change in drought or flood flows." (lines 105-107 of the revised manuscript). We have also better described the regression setup and units in the main text: "We set the regressions with fixed effects for location and use logarithmic-transformed variables." (lines 79-80 of the revised manuscript). Additionally, as recommended in Comment #23 of Reviewer 2, we have clarified in the Methods section how the regressions were set up, their equations, units and transformations.

(14) L99-100: I don't understand the choice of "mean annual per decade". From what I understood from reading the methods, the trends were calculated based on annual data. Therefore, why aren't the values reported as "per year"? Based on how many data points were the regressions calculated?

Reply: The trends are calculated based on annual data, but we transformed it into a "change per decade" so that the results are easier to discuss by not having many decimal points. We have clarified it with:

- We included in the Methods that "We multiplied the trend magnitude by 10 to express it in terms of change per decade." (lines 510-511 of the revised manuscript).
- We included more details on how the trends were computed (see Comment #5 of Referee #2) and included the following sentence in the Methods: "For example, the lower Madeira river in southern Amazonia (gauge ID 15700000, latitude -5.8167, longitude -61.3019) has a drought flow trend of $-0.00588 \text{ mm d}^{-1} \text{ yr}^{-1}$ and a long-term average drought flow of 0.4398 mm d^{-1} , which results in a trend of -13.4% per decade." (lines 513-516 of the revised manuscript).

(15) L105: What is the rationale behind the choice of the hotspots? Why were they picked? This is not clear in the text. Do the results arising from hotspot analysis vary with the choice of different rectangles? In other words, are these hotspots decided based on an intrinsic characteristic? Please try to explain better why the rectangles were placed there and not in other places.

Reply: We have clarified the choice of the hotspots and included a sensitivity analysis respective to the hotspot size and orientation. We have changed the following items in the manuscript:

- We have clarified in the main text why we chose the hotspots: "To interpret our results, we focus on four hotspots of change with distinct streamflow regimes, land management, and in the upstream areas of major South American basins with mounting environmental concerns such as the Amazon, São Francisco, Paraná, Uruguay and Iguazu basins (Extended Data Fig. 2-4)." (lines 118-121 of the revised manuscript).
- We have included more detail in the Methods on why we chose the hotspots and their

characteristics: “We investigate the interannual variability of streamflow and its drivers in four hotspots with mounting environmental concerns (Fig. 2, Extended Data Fig. 2). The selected hotspots are located in the upstream areas of major South American basins with distinct streamflow regimes, land and water management. The Brazilian Highlands hotspot has widespread water-intensive crops with increasing drought and water scarcity issues^{92,93}, which covers the most arid regions upstream of the São Francisco and Paraná basins. The Southern Amazonia and Northern Amazonia hotspots have been under large-scale deforestation with potential hydrometeorological impacts^{41,94,95}, particularly in the south where land cover change is the highest^{39,96}. The Southern Brazil hotspot has been under increasing flooding in recent decades^{69,97}, which covers the upstream areas of the subtropical Uruguay and Iguazu basins.” (lines 586-596 of the revised manuscript).

- We have included the results of a sensitivity analysis respective to the hotspot size and orientation (Fig. R1): “We note that the results are robust to variations in hotspot sizes (by $\pm 20\%$) and orientations (by $\pm 20^\circ$) (not shown).” (lines 596-597 of the revised manuscript).
- We have included a map of the hotspot locations in Fig. 2 so that it is easier to visualize in the main paper.

Original hotspots sizes and orientations

Hotspots 20% smaller

Hotspots 20% larger

Hotspots rotated by 20 degrees

Hotspots rotated by -20 degrees

Figure R1. Streamflow trends and contributing drivers in four hotspots of change for different hotspot sizes and orientations. (Grey boxes) Streamflow trends, with light grey boxes indicating minimum 7-day flows (drought flows), medium grey indicating mean flows, and dark grey

indicating maximum daily flows (flood flows). (Blue boxes) Climatic trends, with light blue boxes indicating minimum 90-day precipitation minus evaporation ($P - E$), medium blue indicating mean $P - E$, and dark blue indicating maximum 14-day $P - E$. (Orange boxes) Water use trends in % of the long-term mean daily streamflow per decade. (Green boxes) Native vegetation cover trends in percentage points of the total area per decade. The boxplots represent the spatial variability of the local trends within each hotspot.

(16) L106: sign, instead of signal?

Reply: We have rewritten the paragraph (see Comment #15).

(17) L107: Maybe once you define, it would be good to use acronyms for the hotspot names such as AS, AN, SB, BH. Maybe a number/letter system? Also, since the hotspot analysis is an explicit part of the results, I recommend including the hotspot bounding boxes in Figure 1 already (with acronyms, or number system, etc.). I see after reading the extended data that acronyms and number system were used. Make sure to follow one method only.

Reply: We have changed all figures to include the acronyms for the hotspot names (instead of numbers or full names) so that it is consistent throughout the manuscript. We have, however, kept the full hotspot names in the text as we find it easier to comprehend.

(18) Figure3: Only here you defined acronyms for the hotspots. It will read better if you introduce them

Reply: We have now defined the acronyms for the hotspots in Fig. 2 and used them in all figures.

(19) L194: Please instruct the reader in the regional climates in Brazil. Is the dry region from May through September a constant across the whole country?

Reply: We have clarified it: “Water abstraction occurs mainly from May to September³⁶ during the dry season in most of central and eastern Brazil, which has caused a substantial reduction in drought flows.” (lines 211-213 of the revised manuscript).

(20) L227-228: Reference needed here.

Reply: We have added two references to the sentence (line 247 of the revised manuscript).

(21) L445: Not sure if I follow the definition of mean annual here. Mean annual should refer to the average of many annual data. How did you calculate the mean annual data per year? I think what was actually done was a mean daily flow per year, or the total annual flow. Please provide a clear description of how that was calculated.

Reply: We have changed it from “mean annual streamflow” to “mean daily streamflow” throughout the manuscript, as better explained in Comments #9 and #10.

(22) L465-467: What is the reason for the choice of 4x4 degree? How sensitive are the results with respect to block dimensions?

Reply: We have now explained the choice of a 4° by 4° block size for the kriging: “The blocks are sized 4° by 4° (approximately 445 by 445 km at the equator), which allows for a robust analysis particularly in the Amazon (where gauge density is the lowest) with on average three gauges in each block.” (lines 527-530 of the revised manuscript).

We have also computed regional trends for blocks sized 1° by 1°, 2° by 2°, 4° by 4°, and 6° by 6° and showed that the results are robust to block size (Fig. R2). Thus, we added the following sentence in the Methods: “Interpolations using block sizes ranging from 1° by 1° to 6° by 6° yield similar results (not shown).” (lines 530-531 of the revised manuscript).

Figure R2. Observed streamflow trends interpolated using varying levels of kriging block sizes. On the left, change in annual minimum 7-day streamflow (drought flows). On the right, change in annual maximum daily streamflow (flood flows). Blue and red indicate increasing and decreasing streamflow respectively (in % change relative to the long-term drought or flood flow, per decade).

(23) L494-496: If panel regression is used to explain the trends, wouldn't it make more sense to have performed trend analysis and panel regression for the same time interval? Does the trend analysis of streamflow-based metrics vary if the 1980-1991 period is excluded?

Reply: We have revised the panel regressions and rewritten their description (see Comment #23 or Reviewer 2), which is now consistent with the trend analysis by using the same time interval (1980-2015). We clarified the description concerning the analysis period: "The panel regressions were computed in two steps. First, we computed the regressions of Equations 1 and 2 for the years 1992 to 2015, which is the period covered by vegetation data. Both regressions had null and non-significant ($P > 0.01$) native vegetation coefficients. Thus, we removed the native vegetation terms and computed the regressions a second time including data from 1980 to 2015 (Fig. 1 inset). The regressions are robust to changes in the analysis period, with similar coefficients for the two time intervals analyzed (1992-2015 and 1980-2015)." (lines 578-584 of the revised manuscript).

Reply to comments by Referee #2

We appreciate the comments of Referee #2 that helped improve the main points of the manuscript. Please, find below our replies to the comments.

The authors' response is printed in blue.

General comment:

(1) I find this paper to be an interesting and valuable contribution to the existing body of literature concerning the compound land and climate drivers of water balance change in Brazil. While the paper is an interesting application of data and methods, some of the methods descriptions are so unclear as to inhibit understanding of the study and interpretation of the results. At present, the findings -- while interesting -- do not appear particularly novel, given the literature documenting similar dynamics in this region. I make specific suggestions below in hopes of helping the authors revise and improve their manuscript.

Major comments:

(2) Abstract: The abstract suggests that this study demonstrates cause and effect: "water use and climate change have amplified..."; "Drying... is due to..."; "Acceleration... is linked to...". The analysis, insofar as I understand it, shows potential associations in time and space between climate and land cover, but does not establish clear causal connections. Therefore, this language should be revised, unless the methodological approaches used in the study can be clarified (see comments below) and shown to support a causal interpretation.

Reply: We have clarified the methodological approaches used in the study, as further explained in the replies to the comments below. We have also replaced "is due to" and "is linked to" by terms that do not establish causality ("is aligned with" and "is related with"). On other hand, we have left "have amplified" because we believe that, with the clarified methodology, there is evidence of causality in the amplification of climate change effects by land management.

(3) 1.58-60 There is no explanation (in the main text or methods) for the selection of the drought (min 7-day) and flood (max daily) flow statistics. Why was the minimum pulled for 7-day periods, but the maximum from daily? Additionally, because these statistics give only one observation per year, the local time series trends (which are then interpolated to the region, and used for subsequent analyses) appear to be calculated based on only 35 observations (1980-2015). I understand there's not much to be done about the limited time range, but it does make me wonder why a more 'stable' statistic wasn't used for the min/max streamflow statistics, like the 5th and 95th, or 10th and 90th percentiles of 7-day or daily streamflow instead. Those percentiles would still be relevant to 'drought' and 'flood flow' conditions, just not the most extreme drought and flood flows. An analysis of true extremes might be justifiable by using the current min/max statistics aggregated over all sites (across sites), but the within-site design of this study (interpolated local trend analysis), wherein trends are fundamentally based on a sample size of 35, doesn't seem well suited to the use of the most extreme observations.

Reply: We use the minimum 7-day streamflow because it is used by Brazilian water management to determine water use. It is also widely used in trend analysis studies worldwide (e.g., Blum et al., 2019; Chagas & Chaffe, 2018; Cigizoglu et al., 2005; Dudley et al., 2020;

Ehsanzadeh & Adamowski, 2010; Fiala et al., 2010; Hodgkins & Dudley, 2011), which enables for a direct comparison with the present study. The rationale behind using the 7-day time scale is to avoid small outliers that could emerge using a 1-day time scale (Smakhtin, 2001). On the other hand, we used maximum daily flow to make it consistent with the minimum 7-day flow and as it is a broadly used index for trend analysis studies (e.g., Blöschl et al., 2017, 2019; Do et al., 2017; Petrow & Merz, 2009; Wasko et al., 2020).

We conducted an alternative analysis using the 5th, 10th, 95th and 90th flow percentiles and found that the trends are close to the 7-day minima and daily maxima (Fig. R3). The Spearman correlations between local trends in minimum 7-day flows with local trends in the 5th and 10th percentiles are 0.94 and 0.93 respectively. Alternatively, the Spearman correlations between local trends in maximum daily flows with local trends in the 95th and 90th percentiles are 0.72 and 0.65 respectively. Therefore, we have decided to keep the 7-day and daily indices in the study and included the following information in the Methods:

- “(i) minimum 7-day streamflow (drought flows), as it is widely used in Brazilian water management and trend analysis worldwide^{69–74}” (lines 487–488 of the revised manuscript).
- “Similarly, changing the minimum 7-day streamflow and maximum daily streamflow by the 5th and 95th flow percentiles yielded similar results, as the Spearman correlations between their local trends are 0.94 and 0.72 respectively.” (lines 500–502 of the revised manuscript).

References:

- Blöschl, G., Hall, J., Parajka, J., Perdigão, R. A. P., Merz, B., Arheimer, B., et al. (2017). Changing climate shifts timing of European floods. *Science*, 357(6351), 588–590. <https://doi.org/10.1126/science.aan2506>
- Blöschl, G., Hall, J., Viglione, A., Perdigão, R. A. P., Parajka, J., Merz, B., et al. (2019). Changing climate both increases and decreases European river floods. *Nature*, 573, 108–111. <https://doi.org/10.1038/s41586-019-1495-6>
- Blum, A. G., Archfield, S. A., Hirsch, R. M., Vogel, R. M., Kiang, J. E., & Dudley, R. W. (2019). Updating estimates of low-streamflow statistics to account for possible trends. *Hydrological Sciences Journal*, 64(12), 1404–1414. <https://doi.org/10.1080/02626667.2019.1655148>
- Chagas, V. B. P., & Chaffe, P. L. B. (2018). The Role of Land Cover in the Propagation of Rainfall Into Streamflow Trends. *Water Resources Research*, 54(9), 5986–6004. <https://doi.org/10.1029/2018WR022947>
- Cigizoglu, H. K., Bayazit, M., & Önöz, B. (2005). Trends in the Maximum, Mean, and Low Flows of Turkish Rivers. *Journal of Hydrometeorology*, 6(3), 280–290. <https://doi.org/10.1175/JHM412.1>
- Do, H. X., Westra, S., & Leonard, M. (2017). A global-scale investigation of trends in annual maximum streamflow. *Journal of Hydrology*, 552, 28–43. <https://doi.org/10.1016/j.jhydrol.2017.06.015>
- Dudley, R. W., Hirsch, R. M., Archfield, S. A., Blum, A. G., & Renard, B. (2020). Low streamflow trends at human-impacted and reference basins in the United States. *Journal of Hydrology*, 580, 124254. <https://doi.org/10.1016/j.jhydrol.2019.124254>
- Ehsanzadeh, E., & Adamowski, K. (2010). Trends in timing of low stream flows in Canada: impact of autocorrelation and long-term persistence. *Hydrological Processes*, 24(8), 970–980. <https://doi.org/10.1002/hyp.7533>
- Fiala, T., Ouarda, T. B. M. J., & Hladný, J. (2010). Evolution of low flows in the Czech

- Republic. *Journal of Hydrology*, 393(3), 206–218.
<https://doi.org/10.1016/j.jhydrol.2010.08.018>
- Hodgkins, G. A., & Dudley, R. W. (2011). Historical summer base flow and stormflow trends for New England rivers. *Water Resources Research*, 47(7).
<https://doi.org/10.1029/2010WR009109>
- Petrow, T., & Merz, B. (2009). Trends in flood magnitude, frequency and seasonality in Germany in the period 1951–2002. *Journal of Hydrology*, 371(1–4), 129–141.
<https://doi.org/10.1016/j.jhydrol.2009.03.024>
- Smakhtin, V. U. (2001). Low flow hydrology: a review. *Journal of Hydrology*, 240(3–4), 147–186. [https://doi.org/10.1016/S0022-1694\(00\)00340-1](https://doi.org/10.1016/S0022-1694(00)00340-1)
- Wasko, C., Nathan, R., & Peel, M. C. (2020). Trends in Global Flood and Streamflow Timing Based on Local Water Year. *Water Resources Research*, 56(8).
<https://doi.org/10.1029/2020WR027233>

Fig R3. Observed streamflow trends in Brazil (1980-2015) according to three different indices of drought flows (left-hand side) and flood flows (right-hand side). Blue and red indicate increasing and decreasing streamflow respectively (in % change relative to the long-term drought or flood flow, per decade).

(4) 1.70-71 I appreciate the concision of this sentence, but in order to understand the analysis and results as presented in the remaining text, it's necessary for the methods to be explained in more depth. I had difficulty understanding the results because I didn't sufficiently understand the study design; I was frequently jumping back and forth between the main text and methods. The analysis and results should be generally understandable from the main text alone, and that is not presently the case. Specifically, the approach used for each component of the analysis (flow

trend detection, attribution of streamflow trends to drivers, focus on hotspots, quadrant analysis, return flow analysis) needs to be summarized in the main text -- either at this location in the text (introduction), or just prior to the discussion of each result. The panel regression model needs to be written out in the Methods at a minimum. At present, there is no model description/equation: I don't know the structure of the model, I don't know how the fixed effects were set up, and therefore I cannot assess to what extent the regression analysis functions in the way it is intended.

Reply: We appreciate the in-depth comment, which has led to a substantial clarification of the methods. We have followed all suggestions, as better explained in the next commentaries. To summarize, we have:

- Clarified which units were used in each portion of the analyses (see Comments #5 and 6).
- Better explained how we computed the drivers (see Comment #20).
- Clarified how we computed the spatial interpolations (see Comments #26 and 31).
- Described the panel regression setup and how to interpret it in the main text (see Comment #8).
- Rewritten the panel regression description in the Methods section to include the equations, model setup, units and transformations (see Comment #23).
- Better described how we delineated the hotspots and how sensitive the results are to changes in hotspot orientation and size (see Comment #9).
- Clarified the quadrant classification motivations and computation (see Comments #13 and 27).

(5) 1.77-onward. The units (% per decade) in drought and flood flows are not clearly defined, and difficult to interpret. For example, the sentence "Regional trends in drought flows range from -37 to +16% per decade..." is unclear in terms of what a trend of "% per decade" actually means. I interpreted this to mean that there was a reduction the size of 37% of some baseline flow in some locations, up to an increase the size of 16% of some baseline flow in other locations, over a 10 year period. In the Methods (1.459) I see that these baseline flows are the average annual flows, and that dividing by the average annual flows was done to "transform it [flows] into units of % change per year". From what I can tell, that is not what the division achieves. Dividing annual drought or flood flow change/year by average annual flow makes the units: 'flow change/year expressed a % of mean annual flow', not "% change per year". I read those as different things. For example, if you saw an annual change in flood flows in some basin equal to 10 m³/year, and the average annual flows in that basin are 100 m³, that annual change of 10 m³/year is equal to 10% of average annual flows -- it's not a 10% change per year. This also creates confusion for interpretation of decadal changes: are the decadal changes just the sum of annual trends: X m³/year (expressed as % of average annual flow) * 10 years? Or, are the decadal changes an actual "percent change" between years: (flow_year 10 - flow_year 1) / flow_year 1 * 100? The same lack of clarity stands for the other variables as well (P, E, water use, forest cover). With respect to water use, it looks like trends there would be change expressed as a percent of *long term* average annual flow (instead of average annual flow). If forest cover is reported in units of % of basin area, then a change in forest cover from year to year, or over a decade, is actually a "percentage point" change -- a change between two

percentages. The specifics of this are really important for interpretation, and at present these specifics are unclear in the results text, in the figures, and in the methods.

Reply: Each trend (except for native vegetation and water use) is expressed in % change relative to the long-term average of the same time series used to calculate the trend. For example, a 10% increase in flood flows per decade would indicate a 10% increase relative to the long-term average flood flow of that river basin. In addition, we multiplied the trends by 10 to transform it from % per year into % per decade so that it is easier to discuss due to fewer decimal points. We have clarified how the trends are computed and their units by:

- We expanded the Methods to include more details on how we calculate the trends: “We multiplied the trend magnitude by 10 to express it in terms of change per decade. The estimated local trend in each series was divided by the long-term average value of its own time series to transform it into units of % change per decade. For example, the lower Madeira river in southern Amazonia (gauge ID 15700000, latitude -5.8167, longitude -61.3019) has a drought flow trend of $-0.00588 \text{ mm d}^{-1} \text{ yr}^{-1}$ and a long-term average drought flow of 0.4398 mm d^{-1} , which results in a trend of -13.4% per decade. There are two exceptions to this transformation: (i) native vegetation cover, for which no transformation was necessary because the data is already in % of the basin area, therefore its trends are expressed in percentage points per decade; and (ii) water use, which was instead divided by the long-term mean daily streamflow because it is a more relevant index to relate to water abstractions.” (lines 510-521 of the revised manuscript).
- We also included the following information in the main text: “Trends in streamflow and their climate drivers are expressed in units of % per decade by dividing each trend by the long-term average value of the same time series.” (lines 105-107 of the revised manuscript).
- We changed the description of the trend units in the caption of Fig. 1 so that it is not confused with mean daily streamflow: “Blue and red indicate increasing and decreasing streamflow respectively (in % change relative to the long-term drought or flood flow, per decade).” (lines 100-102 of the revised manuscript).
- We clarified that water use trends are relative to the long-term mean daily streamflow and that native vegetation trends are expressed in percentage points per decade.
- As recommended by Referee #1 (Comments #9 and #10), we changed from “mean annual flow” to “mean daily flow” throughout the manuscript to differentiate with other terms that refer to the annual time scale and so that it is consistent with the other indices.

(6) Lastly, for the trend analysis, I was unable to locate a location in the paper where the units of streamflow (volumetric, or basin area-averaged depths) are indicated; the regression analysis (1.491) and return flow analysis (1.556-557) discussions in the Methods note units of depth, but it's unclear what's used for the other analysis components. For the return period analysis, you write the use of depths is to address "compatibility of streamflows in catchments of different sizes." There is a need for this same compatibility in the trend and other analyses as well, but it's unclear what streamflow units were used. For example, smaller basins may show more variable flow, potentially creating instability in their drought and flood flow trends relative to larger basins.

Reply: We have included the units in the trend analysis description: “All variables are analyzed in units of mm d^{-1} so that they are independent of basin size, except for native vegetation which is analyzed in % of basin area.” (lines 173-175 of the revised manuscript).

We have also expanded the description of how the trends are calculated, as better described in Comment #5.

(7) 1.99 Are the percent changes here the same as those reported in the main text and in the methods? "percent change of the mean annual value per decade" could mean several things... see comment above.

Reply: Yes. We have clarified the trend units and how they are calculated (see Comment #5) and changed the description of the caption of Fig. 1: "Blue and red indicate increasing and decreasing streamflow respectively (in % change relative to the long-term drought or flood flow, per decade)." (lines 100-102 of the revised manuscript).

(8) 1.100 the panel regression setup, and the interpretation of panel regression coefficients (including units) needs to be explained in the main text and methods before the figure insets can be understood.

Reply: We have clarified the description of the regressions. More specifically:

- We included the regression setup in the main text: "We analyze the links between streamflow changes and their drivers with panel regressions, which allows us to investigate the hydrological variability in space and time in a single framework. We set the regressions with fixed effects for location and use logarithmic-transformed variables." (lines 77-80 of the revised manuscript).
- We clarified how to interpret the regression coefficients in the main text: "A coefficient of 0.5 indicates that a 1% change in a particular driver leads on average to a 0.5% change in drought or flood flows." (lines 105-107 of the revised manuscript).
- We included how the regressions were set up, their equations, units and transformations in the Methods, as better explained in Comment #23.

(9) 1.105 the delineation of the four hotspot regions should be defined more clearly, at a minimum in the Methods. It would help to include a small map of the hotspot locations as well in the main text (along w/ Fig 2), as the hotspot locations only appear in the Extended Data figures. It is not clear how the precise extent and location of these hotspots was determined. I understand the general regional placement, but why the chosen sizes and orientations? In what way was climate, water use, forest cover, and "floods and drought processes" used to determine the locations?

Reply: Referee #1 has suggested similar clarifications (Comment #15), to which we have replied the following.

We have clarified the choice of the hotspots and included a sensitivity analysis respective to the hotspot size and orientation. We have changed the following items in the manuscript:

- We have clarified in the main text why we chose the hotspots: "To interpret our results, we focus on four hotspots of change with distinct streamflow regimes, land management, and in the upstream areas of major South American basins with mounting environmental concerns such as the Amazon, São Francisco, Paraná, Uruguay and Iguazu basins (Extended Data Fig. 2-4)." (lines 118-121 of the revised manuscript).

- We have included more detail in the Methods on why we chose the hotspots and their characteristics: “We investigate the interannual variability of streamflow and its drivers in four hotspots with mounting environmental concerns (Fig. 2, Extended Data Fig. 2). The selected hotspots are located in the upstream areas of major South American basins with distinct streamflow regimes, land and water management. The Brazilian Highlands hotspot has widespread water-intensive crops with increasing drought and water scarcity issues^{92,93}, which covers the most arid regions upstream of the São Francisco and Paraná basins. The Southern Amazonia and Northern Amazonia hotspots have been under large-scale deforestation with potential hydrometeorological impacts^{41,94,95}, particularly in the south where land cover change is the highest^{39,96}. The Southern Brazil hotspot has been under increasing flooding in recent decades^{69,97}, which covers the upstream areas of the subtropical Uruguay and Iguazu basins.” (lines 586-596 of the revised manuscript).
- We have included the results of a sensitivity analysis respective to the hotspot size and orientation (Fig. R1): “We note that the results are robust to variations in hotspot sizes (by $\pm 20\%$) and orientations (by $\pm 20^\circ$) (not shown).” (lines 596-597 of the revised manuscript).
- We have included a map of the hotspot locations in Fig. 2 so that it is easier to visualize in the main paper.

Original hotspots sizes and orientations

Hotspots 20% smaller

Hotspots 20% larger

Hotspots rotated by 20 degrees

Hotspots rotated by -20 degrees

Figure R1. Streamflow trends and contributing drivers in four hotspots of change for different hotspot sizes and orientations. (Grey boxes) Streamflow trends, with light grey boxes indicating minimum 7-day flows (drought flows), medium grey indicating mean flows, and dark grey

indicating maximum daily flows (flood flows). (Blue boxes) Climatic trends, with light blue boxes indicating minimum 90-day precipitation minus evaporation ($P - E$), medium blue indicating mean $P - E$, and dark blue indicating maximum 14-day $P - E$. (Orange boxes) Water use trends in % of the long-term mean daily streamflow per decade. (Green boxes) Native vegetation cover trends in percentage points of the total area per decade. The boxplots represent the spatial variability of the local trends within each hotspot.

(10) 1.107 - 117: Because the analysis has not yet addressed formal attribution (Fig 2 is just a visualization of univariate distributions), consider revising this part to make it clear these are observations of associations: saying "reduction of drought flows is related to decreasing mean $P - E$ " isn't really shown by Fig.2 -- it just looks like they occur at the same time.

Reply: We have clarified that we observed an association between the variables without specifying a cause and effect: "In the Highlands hotspot, a region with intensive agriculture, the reduction of drought flows is aligned with decreasing mean $P - E$ and increasing water use but, from the year 2000 onward, drought flows have become dissociated from mean $P - E$ with a rapid increase in water use (Extended Data Fig. 7)." (lines 125-128 of the revised manuscript).

(11) 1.107-118: This is an interesting discussion, in part because it is at odds to some extent with studies that show deforestation has different and complex effects in this region. Prior literature suggests that removing forest may increase baseflow or mean flow, while not affecting flood flow, with varying interpretations of the role of simultaneous climate variability and change -- deforestation has been shown to potentially mask the effects of climate change on the water balance. Referencing that literature, and discussion of this and other findings in your paper relevant to that literature, is warranted. See:

<https://doi.org/10.5194/hess-21-1455-2017>

<https://doi.org/10.1002/2017GL076526>

<https://doi.org/10.1111/j.1365-2486.2011.02392.x>

<https://doi.org/10.1007/s10584-020-02736-z>

<https://doi.org/10.1016/j.ejrh.2020.100755>

Reply: We have included a discussion of four of the suggested papers, adding the following text: "Annual streamflow to rainfall ratios in three small southern Amazonian basins cultivated with soy beans was found to be twice that of neighboring forested basins, flows in the dry season were lower and those in the wet season were higher⁴⁶ similar to the present study. In contrast, analyses of streamflow in about 50 basins in Amazonia suggest that deforestation has increased low flows, likely because of decreasing transpiration, but without an effect on high flows^{35,47}, indicating that deforestation may potentially mask the effects of climate change on the water balance." (lines 251-258 of the revised manuscript).

The remaining paper (Guimerteau et al., 2017) we have added in the discussion of future changes (line 291 of the revised manuscript).

References:

Guimberteau, M., Ciais, P., Ducharne, A., Boisier, J. P., Dutra Aguiar, A. P., Biemans, H., et al. (2017). Impacts of future deforestation and climate change on the hydrology of the Amazon Basin: a multi-model analysis with a new set of land-cover change scenarios. *Hydrology and Earth System Sciences*, 21(3), 1455–1475. <https://doi.org/10.5194/hess-21-1455-2017>

Hayhoe, S. J., Neill, C., Porder, S., Mchorney, R., Lefebvre, P., Coe, M. T., et al. (2011). Conversion to soy on the Amazonian agricultural frontier increases streamflow without affecting stormflow dynamics. *Global Change Biology*, 17(5), 1821–1833. <https://doi.org/10.1111/j.1365-2486.2011.02392.x>

Heerspink, B. P., Kendall, A. D., Coe, M. T., & Hyndman, D. W. (2020). Trends in streamflow, evapotranspiration, and groundwater storage across the Amazon Basin linked to changing precipitation and land cover. *Journal of Hydrology: Regional Studies*, 32, 100755. <https://doi.org/10.1016/j.ejrh.2020.100755>

Levy, M. C., Lopes, A. V., Cohn, A., Larsen, L. G., & Thompson, S. E. (2018). Land Use Change Increases Streamflow Across the Arc of Deforestation in Brazil. *Geophysical Research Letters*, 45(8), 3520–3530. <https://doi.org/10.1002/2017GL076526>

Rizzo, R., Garcia, A. S., Vilela, V. M. de F. N., Ballester, M. V. R., Neill, C., Victoria, D. C., et al. (2020). Land use changes in Southeastern Amazon and trends in rainfall and water yield of the Xingu River during 1976–2015. *Climatic Change*, 162(3), 1419–1436. <https://doi.org/10.1007/s10584-020-02736-z>

(12) 1.135-137: What estimate of correlation is this referring to: is this temporal (time series) or spatial correlation?

Reply: We have now specified it refers to the spatial correlation of the data shown in Fig. 3b: “Spearman correlation in the spatial variability of regional trends of 0.61, Fig. 3b” (lines 154-155 of the revised manuscript).

(13) 1.138-140: This finding is unclear because the Methods (for the quadrant analysis and other analyses) are insufficiently summarized in the main text. See comment above -- there needs to be a more detailed summary of methods for each analysis component in the main text so that the results can be understood.

Reply: We have rewritten the sentence: “A total of 29% of the study area has been accelerating (Fig. 3b), which is double the expected percentage of a standardized, bivariate normal distribution with a correlation of 0.61 (i.e., the correlation between drought and flood flow trends; Equation 5 in the Supplementary Material). Moreover, 25% and 42% of the study area exhibit wetting and drying trends respectively, whereas 35% would be expected in a standardized, bivariate normal distribution.” (lines 156-162 of the revised manuscript).

(14) 1.177 - 180: this sentence implies that this paper is novel with respect to evaluating compound effects, which is not entirely true (see above comment and suggested papers), although this study is certainly an important contribution to that literature. Again, incorporation of discussion of previous studies of compound effects -- specifically -- here and throughout this section is warranted.

Reply: We have extended the discussion as recommended in Comment #11. In addition, we have expanded the references to include Heerspink et al. (2020) in the first sentence (line 195-198 of the revised manuscript).

References:

Heerspink, B. P., Kendall, A. D., Coe, M. T., & Hyndman, D. W. (2020). Trends in streamflow, evapotranspiration, and groundwater storage across the Amazon Basin linked to changing precipitation and land cover. *Journal of Hydrology: Regional Studies*, 32, 100755. <https://doi.org/10.1016/j.ejrh.2020.100755>

(15) 1.176 - 232 (section): the majority of this section reads like background information or a literature review, not interpretation/discussion of your results specifically. I'd suggest that most of this material be summarized more concisely and moved to the introduction, and then retain only aspects of this information that provide direct context for your specific results here.

Reply: We have left this material in the discussion to more directly relate it to the findings of this paper, but we have made it more specific to the results of the paper to better bring out the connection. The related sentences now read: "One possible explanation for the change is the southward shift of the South American Convergence Zone (SACZ), a major source of precipitation, which has moved away from central Brazil²⁹. The drying trends of Fig. 5 may also be related to a northward displacement of the Intertropical Convergence Zone (ITCZ) which has moved the equatorial precipitation band farther away from northeastern Brazil³⁰." (lines 199-204 of the revised manuscript).

(16) 1.239-240, 253-254: it's not possible to understand the return period results as they are written because the GEV approach/method was not introduced previously in the paper. Again, this method and its purpose, along with all other analysis approach methods, need to be introduced and summarized more clearly earlier in the main text, not just the Methods section at the end. Otherwise, it's not possible to understand the paper.

Reply: We appreciate the concern about the clarity regarding the GEV method, but we would prefer not to include a detailed description in the main text because it is not the focus of the study and would break the flow of the text, as many sentences would be necessary. We have, however, expanded the description of the other methods in the main text (i.e., the trend detection, the units used, the panel regression, and the quadrant classification) as recommended by the previous commentaries.

(17) 1.264-272: this final paragraph is very generic and vague, and adds little insight or value to the paper. Suggest deleting and/or replacing with recommendations that stem from specific findings of this study. For example, it's unclear (in part because of grammar -- see minor comment below) what the suggestion about use of ESMs means, and how it stems from your findings in particular.

Reply: We have modified the middle of the paragraph to enhance its specificity and improve the grammar: "The evidence for the acceleration found here also provides an opportunity for Earth System models to attribute the joint changes in floods and droughts to climate, deforestation and

water use.” (lines 293-295 of the revised manuscript). We still believe that the implications at the end of the paragraph are worth stating.

(18) 1.388-389: The way it is written, I'm not able to tell if you are using data from CAMELS-BR or obtaining data from ANA, but formatting it similar to CAMELS-BR.

Reply: We have now specified in the previous paragraph that “We used daily streamflow data from 886 hydrometric stations obtained from the CAMELS-BR dataset” (lines 419-420 of the revised manuscript). We also clarified that “We removed hydrometric stations with typographic errors and unrealistically large discharges.” (lines 431-432 of the revised manuscript).

(19) 1.424-425: Given that all the listed non-forest land cover categories can have biological (water consumption) properties distinct from deep-rooted forests, particularly in this region, I'd suggest NOT using the term "forest" to describe this aggregation of land types; "natural", "native", or "uncultivated" lands might be more appropriate. Given the prevalence of savanna (shrubland and grassland) vegetation over large swaths of your study region, the term is important.

Reply: We have changed from “forest cover” to “native vegetation cover” throughout the text and in Fig. 2, Fig. 4, Extended Data Fig. 4 and 7 to better convey the presence of native forests, shrublands, grasslands and sparse vegetation.

(20) 1.443: This suggests all P, E, Q statistics (for the trend analysis at least) were computed at the location of the streamflow stations/locations only -- that is, trends in P-E variables were computed at the point/station where streamflow was collected, NOT over the contributing basin area of that streamflow point/station. Is that correct (I'm assuming not, but the paper doesn't make it clear)? The appropriate comparison of streamflow and drivers would come from basin area-averaged estimates of drivers rather than point estimates. If P-E, water use, and forest were NOT computed over the contributing basin areas, then this should be done.

Reply: The information on the contributing basin area was missing from the methods. We have included it in the description of the variables: “The meteorological, native vegetation and water use variables are computed considering the contributing basin area of their respective hydrometric stations.” (lines 492-494 of the revised manuscript).

(21) 1.458-463: see comments above about confusion regarding the definition of % change in all variables.

Reply: We have clarified how the trends are calculated and their units, as better explained in Comment #5.

(22) 1.474: I appreciate the check here, but don't entirely understand if this alleviates the concern, particularly for regression analysis. Can you explain what the concern is regarding inflated variance: what would this do to your trend (or other, i.e. regression) analyses?

Reply: The concern regarding inflated variance due to spatial correlation between stations is that

the interpolated trends could be overestimated in certain parts of the region. We found that the effects of inflated variance are small in the present study. We have clarified the sentence: “To evaluate the possible inflated variance effects on the spatial interpolation due to the spatial correlation between stations, which could lead to an overestimation of regional trends, we repeated the trend interpolation using two subsets of randomly selected stations” (lines 538-541 of the revised manuscript).

(23) 1.486: it is necessary that the regression equations be written out in the text - at a minimum in the Methods, and ideally (in some concise form) in the main text. Stating that you used panel regression with fixed effects similar to other studies (only) is insufficient, and makes it impossible to review the quality of the regression component of this analysis. Specifically: what fixed effects (for time and/or location) were used?

Reply: We have revised the panel regression analysis and rewritten its description. As a result, the coefficients of Fig. 1 (inset) have slightly changed and now indicate a lower (although substantial) impact of water use on drought flows. More specifically, we have:

- Clarified why we chose the fixed-effects model: “We use fixed-effects (for location) regressions as we are mostly interested in analyzing the impacts of variables over time and as indicated by a significant ($P < 0.001$) Hausman specification test⁸⁷.” (lines 553-556 of the revised manuscript).
- Included the regression equations in the Methods (lines 558 and 566 of the revised manuscript).
- Clarified the variable units and how the regression coefficients can be interpreted: “We use logarithms of mm d^{-1} units for all variables except native vegetation as it is already expressed in percent coverage. Therefore, the regression coefficients can be interpreted in relative terms. For example, a 1% change in maximum annual $P - E$ would lead to a $\beta_2\%$ change in flood flows assuming that the remaining independent variables are unchanged.”
- Included additional information about the standardized errors: “We computed the standardized errors of the regression coefficients with robust covariance matrix estimators⁸⁸.” (lines 573-575 of the revised manuscript).
- Included more detail about the panel regression in the main text: “We analyze the links between streamflow changes and their drivers with panel regressions, which allows us to investigate the hydrological variability in space and time in a single framework. We set the regressions with fixed effects for location and use logarithmic-transformed variables.” (lines 77-80 of the revised manuscript).
- Updated the discussion of the results: “Drought trends are driven primarily by changes in mean daily $P - E$, with substantial effects of water use and minimum $P - E$ (Fig. 1a inset).” (lines 110-112 of the revised manuscript).

(24) 1.498: similar results in terms of what? In terms of the coefficients on the independent variables?

Reply: We have rewritten the section regarding the regression analysis and removed the results of traditional multiple regression from the analysis.

(25) 1.498-500: this is insufficiently explained. What are the trends -- from the trend analysis? Per year trends, or the per-decade trends shown in the trend analysis results? What do you mean by similar? Similar in terms of the signs of the coefficients, the magnitude of the coefficients? This is confusing because coefficients from your main regression setup (assuming a simple log-log regression) would be: "a one percent change in flow/year is associated with a one percent change in driver/year"; for the trend regression, based on what I'm reading, the coefficient interpretation would be "a percent change in flow/year² is associated with a one percent change in driver/year²" (in the trend regression case, the coefficient pertains to 'a change in a change in flow', rather than 'a change in flow').

Reply: We have rewritten the section regarding the regression analysis and removed the results of traditional multiple regression from the analysis.

(26) 1.525 It's not clear if for this analysis you also used driver data summarized at the streamflow station locations, or all (interpolated, gridded) locations. In the case of drivers, did you use observations interpolated from the summaries of drivers made at the streamflow station points, or the original gridded data for P-E, water use, forest? The way the main text and methods are written, it's really unclear where you are using original gridded datasets, interpolations made from data summaries done at the streamflow station locations, or data summaries done at the streamflow station location (points). Are the background colors in Extended Data Figs. 3,4,5 from the original gridded data or the interpolations of point-level data summaries?

Reply: For a consistent analysis, we computed the regional trends in the same way for every variable. First we compute local trends (i.e., trends at the hydrometric stations considering their contributing basin areas) and then we obtain regional trends by interpolating the local trends. Therefore, the background colors in Extended Data Figs. 3, 4, 5 are from interpolations of point-level data summaries that include the contributing basin area. We have:

- Clarified the interpolation of the drivers in the Methods: "The regional trends of the drivers are estimated by interpolating the local trends (which considers the contributing basin area of the hydrometric stations) so that it is consistent with the regional streamflow trends." (lines 525-527 of the revised manuscript).
- Clarified what the background colors of Extended Data Figs 3, 4, and 5 indicate in their respective captions: "The background pattern displays regional trends (interpolated from local trends) and the points display local trends (which includes the contributing basin area)."

(27) 1.530-532 It would be helpful to describe this method, and its rationale, in a bit more detail so that readers understand the motivation behind its use and the value of the contribution. For example, it would help to just say Z1 and Z2 were evaluated with respect to drought and flood flows. When discussing rho being set by "regional trends in correlation" - what trends and what correlation: temporal or spatial correlation? From the trend analysis? What component of "this study" does this refer to? In general, I don't understand the intuition behind or motivation for the quadrant analysis approach because it's insufficiently explained.

Reply: We included more precise details on how the quadrant analysis was performed and

included the motivation behind it:

- We included “evaluated with respect to regional trends in drought and flood flows” when introducing Z_1 and Z_2 for the first time (lines 528-529 of the revised manuscript).
- We clarified the correlations and trends used in the quadrant analysis: “Here, we set ρ to 0.61, corresponding to the spatial correlation between the regional trends in drought and flood flows found in the trend analysis in the present study (Fig. 3b).” (lines 535-537 of the revised manuscript).
- We better described the motivation of the quadrant analysis in the main text: “Changes in the extremes may not always be synchronized with changes in mean flows. For example, an increase in mean streamflow combined with an increase in the variance of streamflow could lead to increasing high flows but decreasing low flows. Here, we examine how both flow extremes have changed in a single analysis by classifying the trends into four quadrants (Fig. 3a).” (lines 145-149 of the revised manuscript).

Minor comments:

(28) 1.106 signal should be signals

Reply: We have rewritten the paragraph (see Comment #15 of Reviewer 1).

(29) 1.235 - 262: check grammar throughout these paragraphs: need spaces between instances of "%" and "CI"; delete "have", and add "will" after "events" (1.243); delete "have", and replace "in" with "on" after "damages" (1.245).

Reply: We have checked the grammar and corrected the issues mentioned.

(30) 1.267-268: this is vague.

Reply: We have rewritten the sentence (see Comment #17).

(31) 1.267-269: check grammar. "including land management effects" is unclear - what does this mean?

Reply: We have rewritten the sentence (see Comment #17).

(32) 1.467: why do you suggest this scale? Just explain.

Reply: We have explained our motivations and described the results of a sensitivity analysis as suggested by Referee #1 (Comment #22): “The blocks are sized 4° by 4° (approximately 445 by 445 km at the equator), which allows for a robust analysis particularly in the Amazon (where gauge density is the lowest) with on average three gauges in each block. Interpolations using block sizes ranging from 1° by 1° to 6° by 6° yield similar results (not shown).”

(33) Extended Data Fig. 7: why isn't forest included in panel (a)?

Reply: We have included the forest time series in panel (a) of Extended Data Fig. 7 so that it is consistent with the other panels.

Reviewers' Comments:

Reviewer #1:

Remarks to the Author:

Dear Chagas et al.,

I believe my concerns were addressed and the paper is in good shape for publication. Congratulations!

Antonio Meira